



# Spatiotemporal changes in aerosol properties by hygroscopic growth and impacts on radiative forcing and heating rates during DISCOVER-AQ 2011.

Daniel Pérez-Ramírez[1,2], David N. Whiteman[3], Igor Veselovskii[4], Richard Ferrare[5], Gloria Titos[,1,2], María José Granados-Muñoz[1,2], Guadalupe Sánchez-Hernández[,1,2], and Francisco Navas-Guzmán [2,6].

[1]Applied Physics Department, University of Granada, 18071, Granada, Spain
[2]Andalusian Institute for Earth System Research (IISTA-CEAMA), 18006, Granada, Spain
[3] Atmospheric Sciences Program, Howard University, Washington D.C., 20059, United States.
[4]Physics Instrumentation Center of General Physics Institute, Troitsk, Moscow, Russia
[5]NASA Langley Research Center, Hampton, Virginia, United States.
[6]Federal Office of Meteorology and Climatology, MeteoSwiss, CH-1530, Payerne, Switzerland

*Correspondence to*: Daniel Perez-Ramirez (dperez@ugr.es)

**Abstract.** This work focuses on the characterization of vertically-resolved aerosol hygroscopicity properties and their direct radiative effects through a unique combination of ground-based and airborne remote sensing measurements during the DISCOVER-AQ 2011 field campaign in the Washington D.C. – Baltimore metropolitan area. To that end, we combined measurements from a multiwavelength Raman lidar located at NASA Goddard Space Flight Center and the airborne NASA

Langley HSRL-1 lidar system. In-situ measurements on board the P-3B airplane and ground-based AERONET-DRAGON served to validate and complement quantifications of aerosol hygroscopicity from lidar measurements and also to extend the study both temporally and spatially. The focus here is on the 22nd and 29th of July, 2011 which were very humid days and characterized by a stable atmosphere and increasing relative humidity with height in the planetary boundary layer (PBL). Combined lidar and radiosonde measurements allowed the retrieval of the Hänel hygroscopic growth factor which agreed

with that obtained from airborne in-situ measurements, and also explained the significant increase of extinction and backscattering with height. Airborne measurements also confirmed aerosol hygroscopicity throughout the entire day in the PBL and identified sulfates and water soluble organic carbon as the main species of aerosol particles. The combined Raman and HSRL-1 measurements permitted the inversion for aerosol microphysical properties revealing an increase of particle radius with altitude consistent with hygroscopic growth. Aerosol hygroscopicity was identified as the main reason to explain

aerosol optical depth increases during the day, particularly for fine mode particles. Lidar measurements were used as input to the libRadtram radiative transfer code to obtain vertically-resolved aerosol radiative effects and heating rates under dry and humid conditions, and the results reveal that aerosol hygroscopicity is responsible for larger cooling effects in the shortwave range (7-10 W/m² depending on aerosol load) near the ground, while heating rates produced a warming of 0.12 K/day near the top of PBL where aerosol hygroscopic growth was highest.



## 1 Introduction

Improving our knowledge of atmospheric aerosols is essential to better understanding their role in climate projections because of the uncertainties associated with how atmospheric aerosol particles scatter and absorb solar radiation (direct effect) and how they act as cloud condensation nuclei which affects cloud formation and evolution (Lohman and Feichter, 2005; Haywood and Schulz, 2007). In spite of the advances in understanding aerosol radiative effects, the most recent IPCC model-estimates call for improved understanding of the aerosol indirect effect (Boucher et al., 2013). Although satellite missions and ground-based networks have provided an unprecedented advance in the global knowledge of aerosol optical and microphysical properties, there are still many gaps in the understanding of aerosol changes due to their interaction with water vapor in the atmosphere (Boucher, 2015; Seinfield and Pandis, 2016). Field campaigns are ideal for advancing our understanding of these changes in aerosol properties with water vapor and in how these changes eventually impact direct radiative forcing and cloud formation (Gysel et al., 2007). Ideally these field campaigns include many remote sensing and in-situ instruments because each instrument provides unique information.

Aerosol hygroscopic growth implies changes in aerosol optical and microphysical properties with changing relative humidity (e.g. Titos et al., 2016). Because of the ubiquitous nature of water vapor in Earth´s atmosphere, studying aerosol hygroscopicity is essential for improving our understanding of the role of aerosols in climate (Haywood and Shine, 1995; Pilinis et al., 1995). Hydrophilic particles absorb water vapor and thus increase their size and are become more efficient scatterers (e.g. Zieger et al., 2013). The magnitude of water vapor uptake by an aerosol particle depends directly on their size and chemical composition (Zieger et al., 2013). Hydrophobic aerosols (e.g. mineral particles) show very little variation of their properties with relative humidity (Kotchenruther and Hobbs, 1998; Gasso et al., 2000; Fierz-Schmidhauser et al., 2010a; Titos et al., 2014), while hydrophilic particles (e.g. sulfates, water soluble organic carbonaceous particles) are very sensitive to the uptake of water vapor (e.g. Zieger et al., 2013). Many studies have also demonstrated that hygroscopic growth can accelerate the formation and evolution of haze pollution (e.g. Yang et al., 2015; Chen et al., 2019). Model simulations can be used to evaluate changes in final aerosol direct radiative forcing based on pre-determined models of aerosol hygroscopic growth (e.g. Bian et al., 2009). Actually, Burgos et al. (2020) showed that Earth system global models showed a large diversity in predicting the impact of enhanced relative humidity on aerosol scattering properties, being mainly driven by differences in hygroscopicity parameterizations within the models and model chemistry.

Ground-based in-situ measurements with tandem nephelometers have been widely used to investigate the effect of hygroscopic growth on aerosol scattering properties (e.g. Fierz-Schmidhauser et al., 2010b; Burgos et al., 2019). However, these in-situ measurements are representative only of a few meters above the ground. Tandem nephelometers have also been used onboard aircraft (e.g., Sheridan et al., 2002; Shinozuka et al., 2007) during field campaigns to provide vertically-resolved information on aerosol hygroscopic properties, but such measurements are sparse. Thus, to improve the characterization of aerosol hygroscopicity, the combined use of remote sensing techniques and other meteorological measurements is called for. The use of ground-based sun-photometer from the AERONET network (e.g. Holben et al., 1998)





can provide general insight into aerosol hygroscopicity by analyzing changes in aerosol size distribution with water vapor content (e.g. Schafer et al., 2008) but these characterizations are representative of the column-integrated aerosol. The
combination of lidar with other meteorological measurements is ideal for answering questions about the vertically-resolved aerosol hygroscopicity, but requires the assumption of well-mixed conditions to isolate aerosol hygroscopicity from other aerosol processes (Wulfmeyer and Feingold, 2000, Feingold and Morley, 2003). This technique has been widely used for backscattering lidars (e.g. Granados-Munoz et al., 2015; Haeffelin et al., 2016; Lv et al., 2017; Zhao et al., 2017; Fernandez et al., 2018; Bedoya-Velasquez et al., 2018, 2019; Navas-Guzman et al., 2019; Dowson et al., 2020). However, to obtain
information on how aerosol hygroscopicity affects the aerosol microphysical properties with altitude requires the application of inversion algorithms that use at least measurements of aerosol extinction (α) at two wavelengths (typically 355 and 532 nm) and of backscattering (β) at three wavelengths (typically at 355, 532 and 1064 nm). This approach is generally known as the lidar 3β+2α technique and uses inversion with regularization (e.g. Müller et al., 1999; Veselovskii et al., 2002). Because well-mixed conditions in the boundary layer typically occur during daytime, retrievals of aerosol microphysical properties of
aerosol hygroscopic growth during daytime using Raman lidar are limited due to the difficulty of obtaining independent daytime extinction measurements and to the need of very optimized lidar systems (e.g. Whiteman et al., 2006a,b; Dawson et al., 2020). Thus extrapolations have been done using the closest nighttime measurements (e.g. Veselovskii et al., 2009). However, thanks to the latest technological developments independent vertical profiles during daytime of aerosol extinction and backscattering can be measured using High Spectral Resolution Lidar (e.g. Hair et al., 2009; Burton et al., 2018) and/or
Raman lidar with high power lasers (e.g. Whiteman et al., 2006, 2007) with the use of rotational Raman narrowband filters expanding the use of Raman lidars for daytime measurements (e.g. Ortiz-Amezcua et al., 2019).

Determining aerosol direct radiative effects and heating rates has been challenging in the last decades because their computation requires considerable knowledge of aerosol optical and microphysical properties in order to constrain radiative transfer models (e.g. Ramanathan et al., 2001; Forster et al., 2007). To that end, several recent field campaigns have been
performed to characterize aerosol radiative forcing (e.g. Ramanathan et al., 2007; Mallet et al., 2016). Other studies focused on the characterization of dust radiative effects and heating rates from particular sites (Huan et al., 2009; di Sarra et al., 2011; Perrone et al., 2012; Meloni et al., 2015; Bhawar et al., 2016), and even differentiating between aerosol fine and coarse mode to characterize shortwave and longwave radiative effects (Sicard et al., 2014). All previous studies remarked that to minimize the errors in the computations of aerosol radiative effects it is critical to know the aerosol vertical
distribution (Haywood and Ramaswany, 1998; Guan et al., 2010). In this sense, independent lidar measurements of extinction and backscattering provided by Raman or HSRL systems can minimize errors in radiative forcing computations (e.g. Lolli et al., 2018). To improve the use of lidar data for quantifying vertically resolved aerosol radiative forcing it is necessary to understand how aerosol interacts with other gases in the atmosphere such as water vapor (e.g. Smith et al., 2020; Thorsen et al., 2020), and thus how aerosol hygroscopic growth affects radiative forcing (Rastak et al., 2014).



The Deriving Information on Surface conditions from Column and Vertically Resolved Observations Relevant to Air Quality (DISCOVER-AQ) field campaign conducted in July 2011 in the Washington-Baltimore metropolitan area was a NASA-sponsored field campaign designed to investigate air quality (Crawford and Pickering, 2011) using both airborne and ground-based instrumentation. The airborne assets used in DISCOVER-AQ were the NASA P-3B equipped with a variety of in-situ measurements for aerosol and gas characterization and the Beechcraft King Air UC-12 where the HSRL-1 (Hair et al.,

2009) was installed. Moreover, a unique ground-based set of instrumentation including the multiwavelength Raman lidar system at NASA Goddard Space Flight Center (e.g. Veselovskii et al., 2013, 2015a) was deployed. The combination of airborne and ground-based lidar instruments during DISCOVER-AQ has previously been demonstrated as a useful approach to get accurate 3β+2α measurements used in the inversion by regularization to obtain aerosol microphysical properties (e.g. Sawamura et al., 2014). Moreover, the AERONET-DRAGON (Distributed Regional Aerosol Gridded Observational

Network – Holben et al., 2018) sun photometer network deployment permitted regional aerosol optical depth (AOD) to be characterized at high spatial resolution. AERONET-DRAGON measurements during DISCOVER-AQ 2011 have been used to identify the enhancement of AOD in the presence of cumulus clouds (Eck et al., 2014) but not on how hygroscopic growth can affect the enhancement of AOD over an entire region. Other studies using P-3B data from DISCOVER-AQ 2011 served to identify sulfate and other inorganic compounds as the main aerosol species present during the campaign (e.g. Beyersdorf

et al., 2016) and their possible impact on aerosol hygroscopicity (e.g. Ziemba et al., 2012; Crumeyrolle et al., 2014; Chu et al., 2015). However, there were no closure studies performed on the impact of aerosol hygroscopicity on vertical-profiles of aerosol optical and microphysical properties and their impact on radiative forcing.

The objective of this work is to study how hygroscopic growth affects vertically-resolved aerosol optical and microphysical properties and then to understand how aerosol hygroscopicity affects aerosol loading and radiative forcing. To

that end, the unique dataset of measurements acquired during DISCOVER-AQ 2011 were used with special emphasis on the combination of different multiwavelength lidar systems. Particularly the combination of measurements from the GSFC Raman lidar and the airborne HSRL-1 provided the 3β+2α measurements during daytime. Extensive radiosonde measurements were conducted at the Howard University Beltsville Campus in Beltsville, Maryland, to support DISCOVER-AQ 2011 measurements.

This paper is structured as follow: Section II describes the methodology used. Section III is devoted to the main results while conclusions are given in section IV.

**2 Methodology**

**2.1 DISCOVER-AQ 2011 I: airborne instrumentation**

The DISCOVER-AQ was a multi-year and multi-city NASA mission designed to study air-quality in urban

environments. Here we focus only on the component of the experiment that was performed in the Baltimore-Washington urban region during July 2011 covering a study area approximately from 38.75º to 39.75º N and from 77.75 to 75.25 º W.





The NASA P-3B airplane was equipped with a variety of in-situ instruments to study aerosol and gas-phase particles in the atmosphere. The P-3B flew 14 missions over 29 days in July 2011 including spirals (both up and down) at different reference sites with at least 3 spirals per site. Figure 1a shows an example of the flight track and spirals performed by the P-3B on 29th July. The NASA Langley Aerosol Group Experiment (LARGE - https://science.larc.nasa.gov/large/ ) deployed several in-situ instruments for aerosol characterization: 1) an isokinetic inlet capable of collecting and transmitting particles with diameter smaller than 0.4 µm (McNaughton et al., 2007),  2) an integrating nephelometer (TSI, Inc. model 3563) measuring aerosol scattering coefficients at 450, 550 and 700 nm., 3) a Particle Soot Absorption Photometer (PSAP, Radiance Research) measuring aerosol absorption coefficients at 470, 532 and 660 nm . Combining integrating nephelometer and PSAP measurements the aerosol extinction coefficient and single scattering albedo (ratio of the scattering to the extinction coefficients) are computed using the Angström exponent to obtain scattering coefficient at 532 nm. But all of these measurements are acquired under dry conditions (the sampled air is adjusted to 20% relative humidity – Beyersdorf et al., 2016), and to study the effect of enhanced relative humidity in the scattering coefficients (and adjust the measured value to real atmospheric conditions) another integrating nephelometer is used. This second nephelometer is equipped with a humidification system capable of adjusting relative humidity inside the instrument to 80%. The tandem of nephelometers permits the characterization of the aerosol scattering enhancement factor f(RH) defined as the ratio of aerosol scattering at a certain relative humidity (RH) to the corresponding dry (or reference) scattering coefficient (e.g. Titos et al., 2016). The enhancement factor can be parameterized as a function of RH using the Hänel equation expressed as (Hänel, 1976):

$$f(RH) = \left(\frac{1-RH}{1-RH_{ref}}\right)^{-\gamma} \quad (1)$$

where γ is the hygroscopic growth parameter that quantifies the hygroscopic scattering enhancement.  From the tandem nephelometer measurements and the ambient RH (RH$_{amb}$) it is possible to determine the scattering coefficient at ambient conditions (σ$_{amb}$) as (e.g. Beyersdorf et al., 2016):

$$\sigma_{amb} = \sigma_{dry} \left(\frac{100-RH_{amb}}{80}\right)^{-\gamma} \quad (2)$$

The dependence of absorption with relative humidity is highly uncertain (e.g. Mikhailv et al., 2006; Brem et al., 2012). Nevertheless, for the aerosol types present during DISCOVER-AQ 2011 aerosol absorption was very low thus contributing less than 4% to aerosol extinction with minimum influence on extinction and single scattering albedo (e.g. Ziemba et al., 2013).

The aerosol size distribution was measured by several instruments with each one covering a certain diameter range: the Aerosol Particle Sizer (APS, TSI) for the 0.5 – 20 µm range, the Scanning Mobility Particle Size spectrometer (SMPS - TSI) for the 10-300 nm range, the Ultra-High Sensitivity Aerosol Spectrometer (UHSAS, Droplet Measurement Technologies) for the 60-950 nm range, and the Particle Measure System model LAS-X (LAS, Particle Measuring System, Inc.,) for the range 0.12 – 7.5 µm range. All of these instruments provided aerosol size distribution under dry conditions





which can differ from that in the real atmosphere. On the other hand, water-soluble organic and inorganic species were obtained from a pair of Particle-Into-Liquid Samplers (PILS, Brechtel Manufacturing, Inc.; Weber et al., 2001). The liquid

flow from the second PILS was collected in vials with 3~5 min temporal resolution for offline ion chromatographic analysis of water soluble organic carbon, chloride, nitrate, nitrite, sulfate, sodium, ammonium, potassium, magnesium and calcium mass concentrations (see Beyersdorf et al., 2016 for details).

The NASA Beechcraft King Air UC-12 airplane operated a compact High Spectral Resolution Lidar (HSRL) to obtain vertical profiles of the atmosphere. The HSRL system used spectral sampling of the lidar returned signal to

discriminate the aerosol and molecular components thus permitting independent measurements of aerosol backscattering and extinction coefficients. During DISCOVER-AQ 2011, the HSRL-1 system was capable of obtaining extinction, backscattering and depolarization profiles at 532 nm, and backscattering and depolarization profiles at 1064 nm (Hair et al., 2008; Rodgers et al., 2009). In  total,  13days of flights with two flights/day were performed at a nearly constant altitude of ~8 km. Backscattering and depolarization profiles have a vertical resolution of 30 m and 10 s (~1 km) horizontal resolution,

while aerosol extinction profiles have 300 m vertical and 60 s (~6 km) horizontal resolution. Figure 1b shows the flight track followed by the NASA UC-12 airplane on 29[th] July 2011. Data from both airplanes used in this study are freely available at the NASA Airborne Science Data for Atmospheric Composition (https://www-air.larc.nasa.gov/cgi-bin/ArcView/discover-aq.dc-2011).

### 2.2 DISCOVER-AQ 2011 II: ground-based instrumentation

DISCOVER-AQ 2011 also included a large set of ground-based measurements that complemented those obtained from the aircrafts. Among the most relevant to our work are those provided by the AERONET-DRAGON network (Holben et al., 2018) that provided wide spatial and temporal sun/sky radiometry measurements of spectral AODs and of inversion products such as aerosol size distribution, refractive indices and single scattering albedo (e.g. Dubovik and King, 2000; Dubovik et al., 2000, 2006). For the DISCOVER-AQ campaign, DRAGON consisted of forty-three AERONET sites in the

Washington-Baltimore region covering a region of approximately 125 km long and 40 km wide, following the I-95 corridor from Washington Beltway north to the Maryland-Delaware state line and encompassing both the Baltimore and Washington D.C. suburbs agricultural areas and the Chesapeake Valley. Figure 1c illustrates the spatial distribution of all stations during this field campaign. For the analysis here, we used AERONET Level 2.0 Version 3 data (Giles et al., 2019).

The multiwavelength Mie-Raman lidar deployed at NASA Goddard Space Flight Center (GSFC: 38.99º N, 76.84º

W, 87 m a.s.l.) was unique in providing vertically-resolved aerosol properties. The system consisted of a high power Nd:YAG laser operating at a 50 Hz repetition rate with output powers of approximately 15, 7 and 12 W at 355, 532 and 1064 nm wavelengths. The detector system consisted of a 40 cm aperture Schmidt-Cassegrain telescope operated vertically with a at 0.5 mrad field of view (FOV). The system used interference filters to measure backscattered light at the three laser emission wavelengths plus two additional filters for detecting nitrogen Raman signals at 387 and 607 nm. Complete lidar

overlap occurred at an altitude of approximately 1000 m a.g.l. For each profile, 6000 laser pulses were accumulated which implied a temporal resolution of 2 min. The high power output of the laser permitted Raman measurements at 387 nm during daytime, while for nighttime both Raman measurements were possible. Details of this system can be found in Veselovskii et al., (2013, 2015a,b). In addition, during the field-campaign radiosondes were launched approximately every 4 hours at the Howard University Beltsville Campus (HUBC). HUBC is approximately 8 km distance from GSFC and radiosondes

measurements from HUBC were used to complement Raman lidar measurements by providing vertical profiles of temperature and relative humidity.

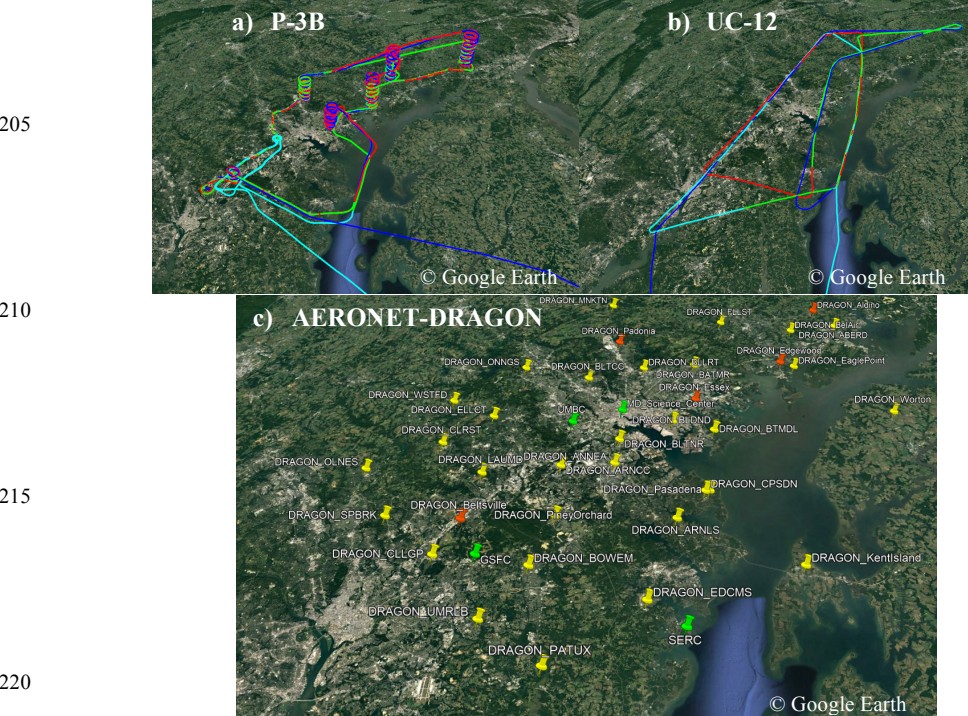

**Figure 1:** For 29th July 2011, (a) flight track for P-3B airplane (b) flight track for UC-12 airplane and (c) AERONET-DRAGON stations. The region is Baltimore-Washington D.C. area covering approximately from 38.75º to 39.75º N and from 77.75 to 75.25 º W. Maps attribution are: Google Landsat/Copernicus Data SIO, NOAA, U.S. Navy, GEBCO. Data LDEO-Columbia, NSF, NOAA

**2.3 Retrieval of vertically-resolved aerosol microphysical properties**

When HSRL-1 flew over GSFC the combination of backscattering at 355 and 1064 nm, and extinction at 355 nm from the ground-based Mie-Raman system with backscattering and extinction at 532 nm from the HSRL-1 system provided



sufficient measurements to perform 3β+2α inversions by solving the ill-posed problem using regularization (e.g. Müller et al., 1999; Veselovskii et al., 2002; Sawamura et al., 2014). But the inversion is underdetermined because of the larger

number of retrieved parameters than input optical data (e.g. Veselovskii et al., 2005; Burton et al., 2016). Therefore, we use an averaging procedure that consists of selecting many solutions in the vicinity of the minimum discrepancy (e.g. Veselovskii et al., 2002). Moreover, constraints in the inversion need to be applied to avoid undesirable solutions (e.g. Whiteman et al., 2018) and the inversion approach uses case-dependent optimized-constraints that limit the ranges of allowed radius and refractive indexes (Perez-Ramirez et al., 2019, 2020). HSRL-1 coincident data with Mie-Raman

measurements are obtained by averaging a maximum of 5 min measurements to guarantee co-location, while Mie-Raman data are averaged every 15 min. The averaging of lidar data is performed to obtain better signal-to-noise ratios, which is particularly important for extinction computations which are the parameters to which the inversion by regularization is most sensitive (Perez-Ramirez et al., 2013).

**2.3 Computations of aerosol radiative forcing and heating rates**

The aerosol radiative effect (ARE) is defined as the perturbation in the solar flux caused by the presence of aerosols in relation to a clean (clear sky, non-aerosol) atmosphere. Thus, aerosol radiative effect within a layer 'z' that is computed as the difference between two sets of radiative fluxes:

$$ARE(z) = (F^{\downarrow}(z)_A - F^{\uparrow}(z)_A ) - (F^{\downarrow}(z)_0 - F^{\uparrow}(z)_0 ) \qquad (3)$$

where $F^{\downarrow}(z)_A$ and $F^{\uparrow}(z)_A$ are the downward and upward fluxes at level z in the presence of aerosols while $F^{\downarrow}(z)_0$ and $F^{\uparrow}(z)_0$ are

the downward and upward fluxes with no aerosols present. The absorption of solar radiation due to aerosols is quantified through the heating rates (HR) that can be defined as (Liou, 2002):

$$HR(z) = \frac{\partial T(z)}{\partial t} = -\frac{1}{\rho c_p}\frac{\Delta F(z)}{\Delta z} = -\frac{1}{\rho c_p}\frac{ARE(z)}{\Delta z} \qquad (4)$$

where $\partial T(z)/\partial t$ is the heating rate at an altitude z, $C_p$ is the specific heat of the air at constant pressure, ρ is the air density and $\Delta F(z)$ is the net flux density divergence or ARE(z) for a given layer of thickness $\Delta z$.

Our computations of AREs and HRs were performed with the LibRadtran radiative transfer model version 2.0.2 (Emde et al., 2016). In the computations with LibRadtran, gas parameterizations from Santa Barbara DISORT Atmospheric Radiative Transfer software (SBDART – Ricchiazzi et al., 1998) and the amounts given for the US standard atmosphere were used. Surface albedos were obtained from AERONET-based surface reflectance validation network (ASRVN – Wang et al., 2011). The large set of data obtained during DISCOVER-AQ allowed the code to be run using real measurements, in

particular: water vapor and temperature profiles from radiosondes launched at HUBC, aerosol single scattering albedo and asymmetry factor obtained from the P-3B aircraft and the AERONET inversions, and most importantly aerosol extinction and backscattering profiles obtained from lidar measurements. Particularly, we used the set of 3β+2α as input to the radiative



transfer model. Outputs from the computations are vertical profiles of ARE and the aerosol heating rate (HR) at 355, 532 and

1064 nm, and also integrated values in the shortwave range between 0.280 and 3 μm. The use of 3β+2α profiles minimizes

the errors in ARE and HR associated with the vertical distribution of aerosols (e.g. Haywood and Ramaswany, 1998; Guan et

al., 2010; Gómez-Amo et al., 2014; Thorsen et al., 2020), and because we use independent measurements by Raman and

HSRL systems this approach should reduce errors associated with the lidar methodology (Lolli et al., 2018).

**3 Results**

**3.1 Impacts of systematic and random uncertainties in determining aerosol hygrosocpicity parameter from lidar**
**derived parameters**

The approach to identify the conditions for studying aerosol hygroscopicity in the atmospheric aerosol profile consists of

identifying well-mixed layers that guarantee the same type of aerosol with altitude. Under these conditions, enhancement of

aerosol backscattering or extinction with relative humidity can be associated directly with water vapor uptake by aerosol

particles (e.g. Ferrare et al., 1998; Wulfmeyer and Feingold, 2000). To that end, the approach assumes that well-mixed layers

are those with constant water vapor mixing ratio with altitude (e.g. Veselovskii et al., 2009). In such circumstances an

increase of relative humidity with altitude is typically observed, and the Hanël equation, including lidar backscattering

coefficients, is given as:

$$f(RH) = \frac{\beta(RH)}{\beta(RH_{ref})} = \left(\frac{100-RH}{100-RH_{ref}}\right)^{-\gamma} \qquad (5)$$

where $RH_{ref}$ and $\beta_{ref}$ are the relative humidity and aerosol backscattering coefficients at the lowest altitude of lidar

measurements, respectively. Taking logarithms of both sides of Eq.5 yields:

$$log\left(\frac{\beta(RH)}{\beta(RH_{ref})}\right) = -\gamma log\left(\frac{100-RH}{100-RH_{ref}}\right) \qquad (6)$$

And now a linear regression of the measurements of β(z) and RH(z) at altitudes exhibiting aerosol hygroscopic

growth provides the hygroscopic growth parameter 'γ' as the slope of the best fit equation.

The impact of systematic and random uncertainties in relative humidity on the computation of the hygroscopic

growth parameter 'γ' is studied here to determine the uncertainties in γ that are obtained from lidar measurements and to

understand better how γ obtained from lidar measurements compares with values obtained from other measurements. First,

the effects of random uncertainties in estimations of γ are studied through simulations where uncertainties in relative

humidity were generated from a Gaussian distribution centered at zero with width equal to the random uncertainty desired

(1, 2, 3, 5, 7, 8, 10, 20, 25 and 30%). We note that random uncertainties in relative humidity with radiosondes are ~ ±5%

(e.g. Milosevich et al., 2009) while these obtained by lidar can be as large as ±30% or more depending on lidar measurement

statistics (e.g. Whiteman et al., 2007). First, Eq. 5 is used with γ equal to 0.4, 0.6, 0.8, 1.0 and 1.2 to calculate the value of





f(RH) with no uncertainty in relative humidity which is allowed to range from 20-100%. Later, for each range of random uncertainty a new set of relative humidities is calculated from the Gaussian distributions that allows a new γ' to be computed. Relative differences in the hygroscopicity parameter are calculated as $(\gamma - \gamma')/\gamma$. Figure 2a show the relative

difference in hygroscopicity parameter versus random uncertainties in relative humidity. The mean differences are shown with errors bars indicating the standard deviations of the spread in the results. Figure 2a clearly indicates that relative differences in γ decrease until reaching approximately constant values of -20% for relative uncertainties larger than 10%. Actually, for random uncertainties below 5%, typical of radiosondes, uncertainties in γ are below -15%. We note that in all computations random uncertainties were positive.


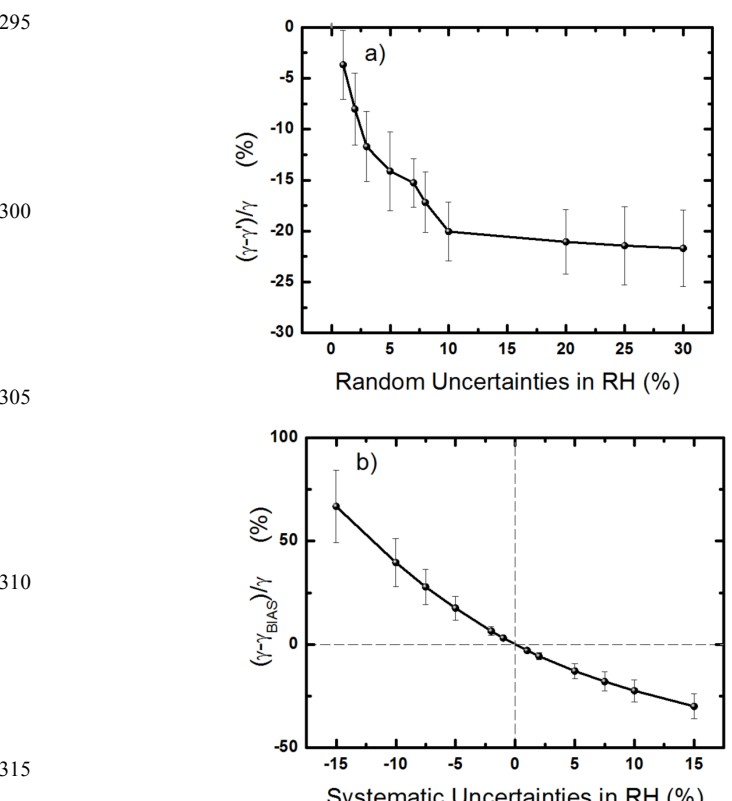





**Figure 2:** Sensitivity analyses of the hygroscopic growth parameter 'γ' as function of (a) random errors in the relative humidity and (b) systematic errors in relative humidity

To evaluate the effects of systematic uncertainties in relative humidity on the calculation of γ we again performed
simulations but applied a fixed bias over the entire range of relative humidity. These biases are associated with systematic





uncertainties which vary from -15% to +15% in the simulation. With relative humidity affected by bias a new hygroscopicity parameter, $\gamma_{BIAS}$, is computed and serves to evaluate the relative differences $(\gamma - \gamma_{BIAS})/\gamma$. Figure 2b shows mean values and standard deviations of the relative differences. Fig. 2b clearly shows a decrease in relative differences from maximum of ~65% for biases of -15 % to a minimum close to -30% for biases of ~15 %. For biases within ±5%, which is what radiosondes can possess due to dry biases (Milosevich et al., 2009), the uncertainties in $\gamma$ in are in the range ±15%. Based on all these results we can conclude that the assessment of gamma is considerably more sensitive to systematic uncertainties in RH than random uncertainties.

**3.2 Characterization and validation of aerosol hygroscopicity computed with lidar measurements.**

During DISCOVER-AQ typical summertime mid-Atlantic conditions of hot and humid days were present. This weather coupled with local emissions and transport of anthropogenic aerosol produced ideal conditions for studying aerosol hygroscopicity. Because of the limited availability of Mie-Raman ground-based measurements during DISCOVER-AQ 2011, only two particular days were identified for studying aerosol hygroscopicity with lidar measurements.

The first day was July 29, 2011 which offered a 10 hours record of daytime ground-based Mie-Raman measurements. Backward trajectories computed by the HYSPLIT model (Stein et al., 2015) revealed very stable atmospheric conditions on this day with air-masses predominantly coming from the Ohio river valley. The temporal evolution of atmospheric profiles of water vapor mixing ratio (Figure 3a) revealed a well-mixed lower atmosphere at 20:00 UTC with constant water vapor mixing ratio of ~ 12.0 g/Kg from the ground up to ~2.2 km. In this layer, relative humidity increased from ~ 33 % at the surface to ~75 % at 2.2 km as shown in Figure 3b. During the rest of the days well-mixed conditions are not present as revealed the variability of water vapor mixing ratio with height in the first kilometers above the ground.

The other case study occurred on July 22, 2011 although for this case the Mie-Raman lidar measurements were sparse extending from 18:00 UTC to 19:00 UTC approximately. According to HYSPLIT, airmasses again had their origin in the Ohio river valley. On this day at 18:15 UTC water vapor mixing ratio profiles from HUBC radiosonde measurements (Fig. 3c) revealed two very stable atmospheric layers, the first from the ground up to approximately 1.8 km with constant water vapor mixing ratio of ~ 15.5 g/Kg, and the second from 3 to 4.5 km with a much lower water vapor mixing ratio of ~ 3.5 g/Kg. Both layers were characterized by an increase of relative humidity from the bottom to the top of the layers (Fig. 3d), although only the first layer covered a range of relative humidity large enough to study aerosol hygroscopic growth (from 40 to 80% approximately). These patterns of water vapor mixing ratio and relative humidity variation with height seem to start early as suggested by the data at 13:57 UTC. Stable conditions are also observed at 03:51 UTC, but not for the rest of the hours when large variability of relative humidity is observed near the surface.

Figure 4 shows temporal evolution of $\alpha(355)$ from multiwavelength Raman lidar measurements performed at GSFC on July 29, 2011 using the Klett method (Klett, 1985) with extinction-to-backscattering ratio, otherwise known as the lidar ratio (LR), of 85 which was computed from correlative AERONET inversions. The use of fixed lidar ratio and the

hypotheses in its computation induce errors in backscattering retrievals, the use of Klett method is used here for the visualization of data with high temporal resolution (1 min). In Figure 4, times when HSRL-1 flew over GSFC are indicated

with white lines, while dotted lines indicate when the P-3B airplanes performed spirals over the HUBC. Times of coincident radiosondes are shown with dashed white lines.

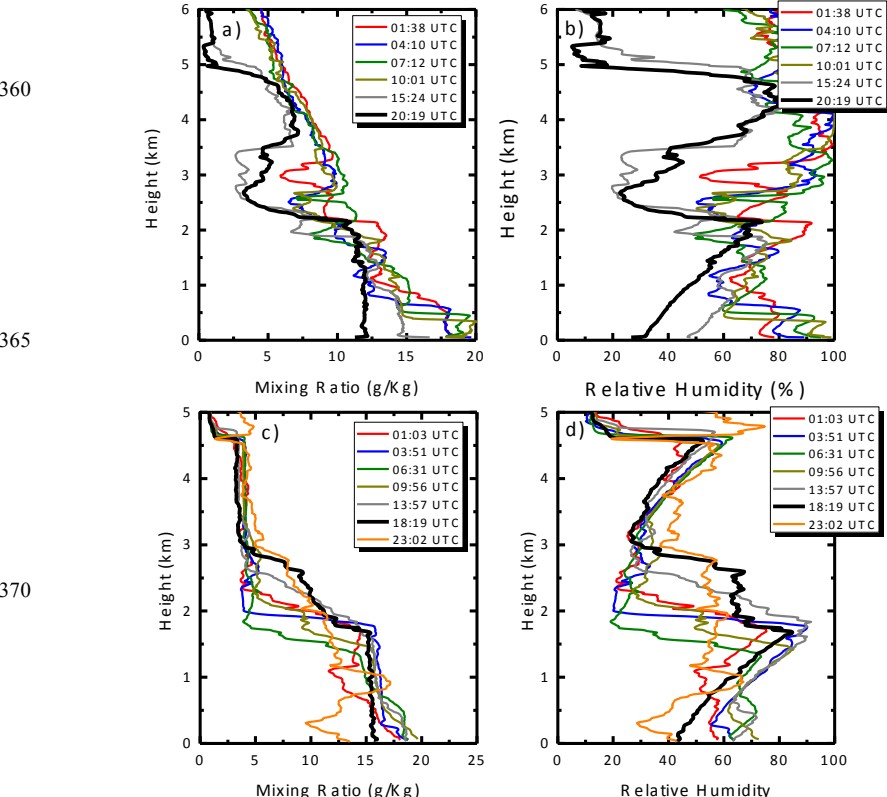

**Figure 3:** Water vapor mixing ratio and relative humidity profiles obtained from radiosondes launched at HUBC on (a-b) 29[th] July 2011 and on (c-d) 22[nd] July 2011.

Figure 4 indicates that most aerosols are within the planetary boundary layer (PBL) with sporadic and thin de-coupled layers between 4-5 km that are associated with transport of biomass-burning from fires active in the western U.S

(https://firms.modaps.eosdis.nasa.gov/). But the most important feature to note is the evolution of aerosols within the PBL throughout the measurement period. Early in the morning de-coupled aerosol layers are observed (note that local time is

UTC-4). At approximately 16:00 UTC it is observed that as the PBL height increases intense red areas corresponding to convective clouds formed at the top of the boundary layer. After approximately 19:40 UTC there were clear skies and the most remarkable feature is the increase of aerosol extinction with altitude over approximately 2.5 hours and the combination

of Mie-Raman with HSRL-1 measurements serve to get 3β+2α measurements and study the possible influence of aerosol hygroscopicity based on lidar measurements.

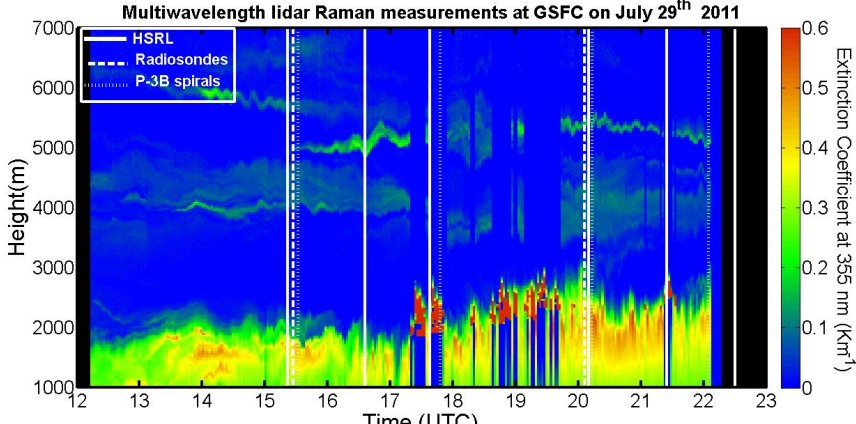

**Figure 4:** Time-evolution of aerosol extinction coefficient at 355 nm from GSFC Raman lidar measurements on 29[th] July 2011. Extinction data are computed using the Klett method with a lidar ratio of 85 sr. Vertical resolution is 7.5 m. Also noted in the plot are overpasses by
the HSRL-1 system and correlative spirals by the P-3B airplane with in-situ instrumentation close to GSFC-HUBC region. Dotted lines represent times when radiosondes were launched.

Figure 5 shows lidar derived parameters for two selected case studies when 3β+2α measurements were possible for studying aerosol hygroscopicity. The two selected cases are on July 29 at 20:00 UTC (Figure 5 a-b) and July 22 at 18:35 UTC (Figure 5c-d). Data are shown for the region above 1 km a.g.l where Raman lidar measurements had complete overlap.

Specifically, Figure 5 also shows correlative measurements of aerosol backscattering at 355, 532 and 1064 nm between ground-based Mie-Raman lidar and airborne HSRL-1. For aerosol backscattering coefficient at 532 nm the reference measurements are from HSRL-1 because it obtains independent extinction and backscattering measurements at this wavelength, while at 1064 nm we present just a comparison because both lidar systems are only capable of obtaining backscattering measurements at this wavelength. On the other hand, the Mie-Raman system provided independent aerosol

extinction and backscattering coefficients at 355 nm using the Raman measurements at 387 nm, while for 532 and 1064 nm the Klett method is used with LR of 65 and 30 on July, 29 and of 55 and 40 on July, 22 for 532 and 1064 nm, respectively. For both days, Figure 5 indicates that both instruments reveal very similar values and vertical patterns in backscattering and extinction, which illustrates the consistency of the measurements and lends confidence to their use as a set of 3β+2α measurements.





Figure 5a-b shows that at 20:00 UTC on July 29 both backscattering and extinction increase with altitude in the planetary boundary layer. The constant water vapor mixing ratio and the increase of relative humidity with altitude at 20:19 UTC (Figure 3a-b) make this set of 3β+2α measurements ideal for studying aerosol hygroscopicity. For July 22 at 18:35 UTC again is observed an increase of extinction and backscattering with height, and the closest radiosonde was at 18:15 UTC (Figure 3 c-d) indicating well-mixed conditions that were therefore good for studying aerosol hygroscopic growth

during the day. Unfortunately, there were only ~2 hours of multiwavelength ground-based Raman lidar on July, 22 and measurements on this day will serve to complement the large set of measurements for July 29.

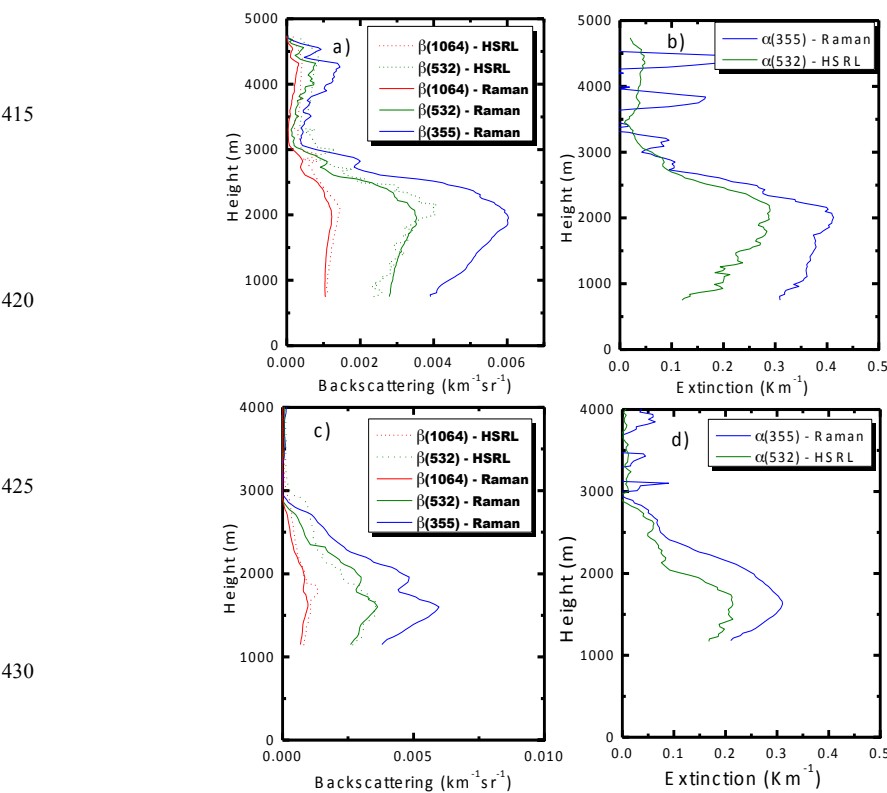

**Figure 5:** Vertical profiles of backscattering (β) and extinction (α) measurements obtained by ground-based Raman lidar and airborne HSRL-1 at NASA Goddard Space Flight Center on (a-b) 29th July 2011 at 20:00 UTC and on (c-d) 22nd July 2011 at 18:35 UTC. The ground-based Mie-Raman measurements provided independent measurements of backscattering and extinction coefficients at 355 nm, while the airborne HSRL-1 system provide those for 532 nm. For 1064 nm both instruments provide only backscattering measurements.





Figure 6 shows f(RH) – computed as $\beta(RH)/\beta(RH_{ref})$ - as a function of relative humidity for the two study cases on July 22 and 29, 2011. The study regions are ones in the planetary boundary layer where backscattering increases with altitude (Figure 5) and where water vapor mixing ratio is found to be constant in the layer thus leading to an increase of relative humidity with altitude (Figure 3). The reference values $RH_{ref}$ needed to compute f(RH) in Eq. 5 were selected at the lowest altitudes where lidar measurements were reliable. The linear fits to Eq. 6 provided the hygrocopicity parameter $\gamma$. We

note that linear fits are unweighted with no dependences on the uncertainty of lidar derived parameters. The results of the linear fits are summarized in Table 1 for the different days and wavelengths. Table 1 also includes the computations of the hygroscopicity parameter $\gamma$ with their corresponding uncertainties of $\sim \pm15\%$ that is typical for type of radiosondes used (see section 3.2), and the computed f(RH)* using RH of 80 and 20% as humid and dry values that are used in the computations by the tandem nephelometers. High correlations are found between aerosol backscattering and relative humidity ($R^2 > 0.83$)

implying that aerosol hygroscopic growth is the likely explanation for the increase in aerosol backscattering with height. Differences between both days are found for 355 and 1064 nm, but, within the uncertainties, the same value of gamma is obtained at 532. Spectral dependences that are larger than the uncertainties were observed for the computed $\gamma$ values, with larger values at 355 nm (0.46 - 0.66) than at 1064 nm (0.31 -0.37). This suggests that changes in scattering by aerosol hygroscopicity are sensitive to wavelength, which agrees with studies at other locations (e.g. Navas-Guzman et al., 2019).

| | | Linear fit parameters from lidar measurements | | | Computed Hygroscopicity Parameters | |
|---|---|---|---|---|---|---|
| | Wavelength (nm) | Slope | Intercept | $R^2$ | f(RH)* | $\gamma$ |
| **29th July 2011** | 355 | -0.46 | 0.01 | 0.910 | 1.89 | $0.46 \pm 0.07$ |
| | 532 | -0.39 | 0.04 | 0.886 | 1.71 | $0.39 \pm 0.06$ |
| | 1064 | -0.31 | 0.01 | 0.850 | 1.54 | $0.31 \pm 0.05$ |
| **22nd July 2011** | 355 | -0.65 | 0.07 | 0.895 | 2.46 | $0.65 \pm 0.10$ |
| | 532 | -0.38 | 0.05 | 0.832 | 1.70 | $0.38 \pm 0.06$ |
| | 1064 | -0.37 | 0.05 | 0.837 | 1.67 | $0.37 \pm 0.06$ |

**Table 1:** Results of the linear fits of log(f(RH)) versus log((1-RH)/(1/$RH_{ref}$)). The computation of f(RH)* was done using Eq. 5 with RH = 80% and $RH_{ref}$ = 20% for intercomparisons with tandem of nephelometers.

        During the time that lidar data from Figure 5 were acquired, the P-3B airplane performed spirals up and down over the HUBC location. Figure 7 shows P-3B airborne measurements of hygroscopicity parameters $\gamma$ and f(RH). The scattering

coefficient at 532 nm ($\sigma_{dry}$ - computed using Angström law) and the number of particles obtained by the UHSAS ($N_{UHSAS}$) instrument are also shown, both being computed at dry conditions. The most remarkable result from Figure 7 is that for these four parameters there are no significant differences within the planetary boundary layer indicating well-mixed conditions. Above these limits (~2200 m on July 29 and ~1800 on July 22), the hygroscopicity parameters are also stable although very noisy due to the considerably lower aerosol loads. Also, $\sigma_{dry}$ and $N_{UHSAS}$ can be seen to decrease above the boundary layer

from their approximately constant values below it. The only exception is on July 22 during the spiral-up when a sharp increase in $\sigma_{dry}$ and $N_{UHSAS}$ occurs near the top of the boundary layer, possibly due to accumulation of pollutants.










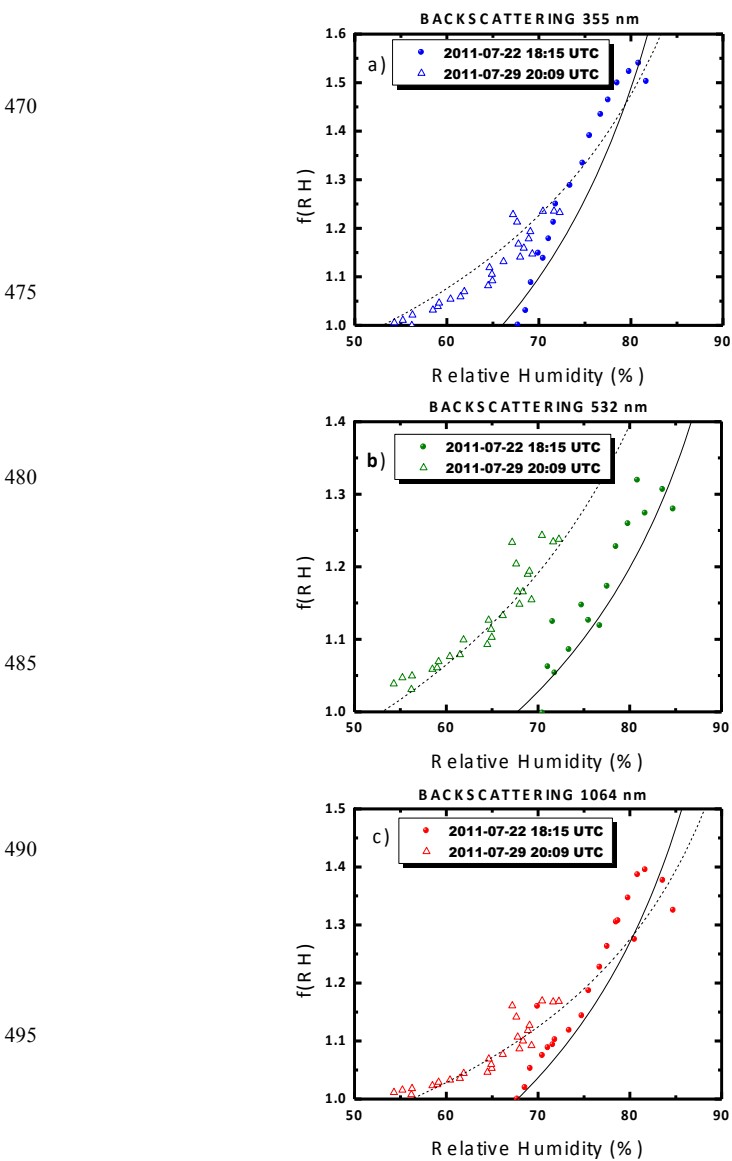

**Figure 6:** Humidigrams of f(RH) versus relative humidity for the aerosol backscattering data obtained on 29th July 2011 and on 22nd July 2011. Data are presented at the wavelengths of (a) 355 nm, (b) 532 nm and (c) 1064 nm.

**Figure 7:** Vertical profiles of hygroscopicity parameter γ and F(RH), scattering coefficient at dry conditions ($\sigma_{dry}$) and number of particles from UHSAS ($N_{UHSAS}$) acquired from the P-3B airplane over GSFC-HUBC region on (a) 29th July spiral-down, (b) 29th July spiral-up, (c) 22nd July spiral-down and (d) 22nd July spiral-up





We computed the mean aerosol parameters obtained by the P-3B in the planetary boundary layer, with the optical results being summarized in Table 2 (hygroscopicity parameters γ and f(RH), absorption ($\sigma_{abs,dry}$) and scattering ($\sigma_{scat,dry}$) coefficients and single scattering albedo (SSA) for dry conditions, and scattering coefficient for ambient conditions ($\sigma_{scat,amb}$)). All parameters are given at the reference wavelength of 532 nm. Similarly, results of microphysical parameters are summarized in Table 3 (number and volume of particles computed by SMPS, LAS, UHSAS and APS instruments). The very low standard deviations for all optical and microphysical parameters - except for ($\sigma_{scat,amb}$) - indicate well-mixed

conditions in the planetary boundary layer.

| | 2011/07/29 20:09 UTC (500-1900 m ) | | 2011/07/22 18:35 UTC (500-1500 m ) | |
|---|---|---|---|---|
| | Spiral-Down | Spiral-Up | Spiral-Down | Spiral-Up |
| $\sigma_{abs}$ (Mm$^{-1}$) | 0.4 ± 0.4 | 0.5 ± 0.3 | 2.12± 1.06 | 1.65 ± 0.45 |
| $\sigma_{scat,dry}$ (Mm$^{-1}$) | 166 ± 10 | 186 ± 3 | 121± 13 | 119 ± 4 |
| $\sigma_{scat,amb}$ (Mm$^{-1}$) | 288 ± 28 | 334 ± 7 | 212± 26 | 213 ± 7 |
| SSA | 0.99 ± 0.01 | 1.00 ± 0.01 | 0.99± 0.01 | 0.99 ± 0.01 |
| γ | 0.39 ± 0.03 | 0.43 ± 0.02 | 0.40 ± 0.02 | 0.42 ± 0.02 |
| f(RH) | 1.73 ± 0.08 | 1.80 ± 0.04 | 1.74 ± 0.06 | 1.80 ± 0.04 |

**Table 2:** Mean values in the planetary boundary layers of hygroscopicity parameters γ and F(RH), absorption ($\sigma_{abs,dry}$) and scattering ($\sigma_{scat,dry}$) coefficients and single scattering albedo (SSA) for dry conditions, and scattering coefficient for ambient conditions ($\sigma_{scat,amb}$). Reference wavelength for all aerosol optical parameters is 532 nm.

| | 2011/07/29 20:09 UTC (500-1900 m ) | | 2011/07/22 18:35 UTC (500-1500 m ) | |
|---|---|---|---|---|
| | Spiral-Down | Spiral-Up | Spiral-Down | Spiral-Up |
| N – SMPS (cm$^{-3}$) | 2680 ± 470 | 2394 ± 270 | 1784± 800 | 2350 ± 300 |
| V – SMPS (µm$^3$/cm$^3$) | 5.1 ± 0.4 | 5.4 ± 0.6 | 3.5 ± 1.3 | 4.9 ± 0.5 |
| N – UHSAS (cm$^{-3}$) | 1899 ± 140 | 1860 ± 97 | 2280± 510 | 2260 ± 200 |
| V – UHSAS (µm$^3$/cm$^3$) | 13.3 ± 1.7 | 13.6 ± 0.9 | 11.3 ± 1.6 | 10.7 ± 1.0 |
| N –LAS (cm$^{-3}$) | 1652 ± 150 | 1654 ± 107 | 1980 ± 370 | 1814 ± 170 |
| V – LAS (µm$^3$/cm$^3$) | 13.1 ± 1.5 | 13.1 ± 0.9 | 11.3 ± 1.6 | 9.9 ± 0.9 |
| N – APS (cm$^{-3}$) | 0.37 ± 0.18 | 0.41 ± 0.19 | 0.41 ± 0.18 | 0.44 ± 0.21 |
| V – APS (µm$^3$/cm$^3$) | 0.05 ± 0.04 | 0.06 ± 0.05 | 0.06 ± 0.04 | 0.07 ± 0.05 |

**Table 3:** Mean values in the planetary boundary layers of total number of particles (N) and volume of particles (V) obtained by SMPS, UHSAS, LAS and APS instruments. Data are representative for dry conditions.

The most important finding is that comparing γ parameters from P-3B airplane in Table 2 with those obtained in Table 1 from lidar measurements (γ values of ~0.39 at 532 nm derived from lidar parameters and of 0.39-0.43 from in-situ

airplane measurements) we observe very similar results with the differences below 5%. The comparison for f(RH) between both methodologies also shows very good agreement. These results and the constant values of γ and f(RH) by the P-3B measurements serve as a validation of the methodology for characterizing aerosol hygroscopic effects on the aerosol vertical profiles using lidar methodology. The fact that the total number and volume of particles remains constant through the boundary layer supports the hypothesis of the same aerosol type in dry conditions with well-mixed atmospheric conditions.

This last point agrees also with the constant $\sigma_{scat,dry}$, while the $\sigma_{scat,amb}$ as computed with Eq. 1 yields profiles very similar to





those of Figure 5. Therefore, we conclude that aerosol hygroscopicity is the main cause of vertical changes in aerosol backscattering. The increase of extinction with height observed in Figure 5 can be also associated with aerosol hygroscopic growth assuming the same Hänel parameters obtained.

Table 2 reveals that aerosol hygroscopicity is very similar between the two days studied with values of γ and f(RH)
that agree to within 5%. For both days aerosol absorption was found to be negligible with SSA very close to 1. Also, essentially no variability was found in the absorption vertical profiles by the P-3B measurements (graphs not shown for brevity). Changes in scattering and extinction coefficients between both days can be explained by the different aerosol loads as indicated by the total number of particles shown. To investigate what aerosol types might have been present, Table 4 shows the mean mass of the different species measured by the PILS instrument. We note that for some species there are no
measurements because their amount in the atmosphere was below the detection limit of the instrument and thus negligible. Also, the large integration time for obtaining the chemical composition did not allow the retrieval of vertical profiles at the resolution similar to that in Figure 7. Indeed, the low standard deviations for each of the species again suggests a well-mixed layer below the top of the planetary boundary layer.

| | 2011/07/29  20:09 UTC  (500-1900 m ) | | 2011/07/22  18:35  UTC  (500-1500 m ) | |
|---|---|---|---|---|
| | Spiral-Down | Spiral-Up | Spiral-Down | Spiral-Up |
| Chloride ($\mu g/m^3$) | NA | 0.048 | NA | NA |
| Nitrite ($\mu g/m^3$) | 0.018 | 0.055 | 0.016± 0.003 | 0.007 |
| Nitrate ($\mu g/m^3$) | 0.037± 0.004 | 0.032± 0.004 | 0.38± 0.15 | 0.227 |
| Sulfate ($\mu g/m^3$) | 8.5± 2.2 | 9.7± 0.6 | 6.2± 0.3 | 5.7± 0.1 |
| Sodium ($\mu g/m^3$) | NA | 0.057 | 0.017± 0.003 | NA |
| Ammonium ($\mu g/m^3$) | 2.5± 0.4 | 4.2± 0.9 | 2.5± 0.2 | 2.2± 0.1 |
| Potassium ($\mu g/m^3$) | NA | NA | NA | NA |
| Magnesium ($\mu g/m^3$) | NA | NA | NA | NA |
| Calcium ($\mu g/m^3$) | NA | NA | NA | NA |
| Black Carbon Mass ($\mu g/m^3$) | 0.25± 0.08 | 0.25± 0.06 | 0.29± 0.10 | 0.27± 0.06 |
| Water Soluble Organic Carbon Mass ($\mu g/m^3$) | 5.19± 0.25 | 6.09± 0.17 | 3.7± 0.3 | 4.3± 0.1 |
| Total | 16.57 | 20.432 | 13.10 | 12.70 |

**Table 4:** Mean values of total mass of the different species that form aerosol particles in the planetary boundary layers.

Table 4 reveals that sulfate is the predominant specie for both days with a percentage ranging between 45-52% for the two days. Carbonaceous species are the second most prevalent comprising 30-36% of the total mass with water soluble organic carbon accounting for at least 93% of the total carbon mass. Sulfate and water soluble organic carbon are hydrophilic and explain the large effect of aerosol hygroscopicity on aerosol properties over the study region. Other important species that were present are ammonium with a percentage of 15-20%. The rest of the species are generally negligible with nitrates
being only of interest at 3% of total mass on July 22.

To further study the present cases, retrievals of aerosol microphysical properties from 3β+2α lidar measurements were made using case-dependent optimized constraints (Perez-Ramirez et al., 2019), which for the data of Figure 5 were of low absorption ($m_{i,max}$= 0.01, $m_{r,max}$= 1.45) and fine mode predominance ($r_{max}$ = 2 μm). Figure 8 shows the main results of bulk parameters (effective radius ($r_{eff}$), aerosol volume (V) and number concentrations (N)) and imaginary refractive index,

both for July 29 (Figure 8 a-d) and July 22 (Figure 8 e-f). Because of the use of case-dependent optimized-constraints, results are representative of fine mode aerosols and uncertainties are of ~25% for $r_{eff}$ and V and of ~100% for N (Pérez-Ramírez et al., 2013) while for mr they are ± 0.05 (Perez-Ramirez et al., 2020). Figure 9 shows particle volume size distributions for different altitudes both below and above the planetary boundary layer, but we note that particle size distributions obtained by the stand-alone 3β+2α lidar inversion can possess significant errors of up to 100% (e.g. Veselovskii et al., 2004).


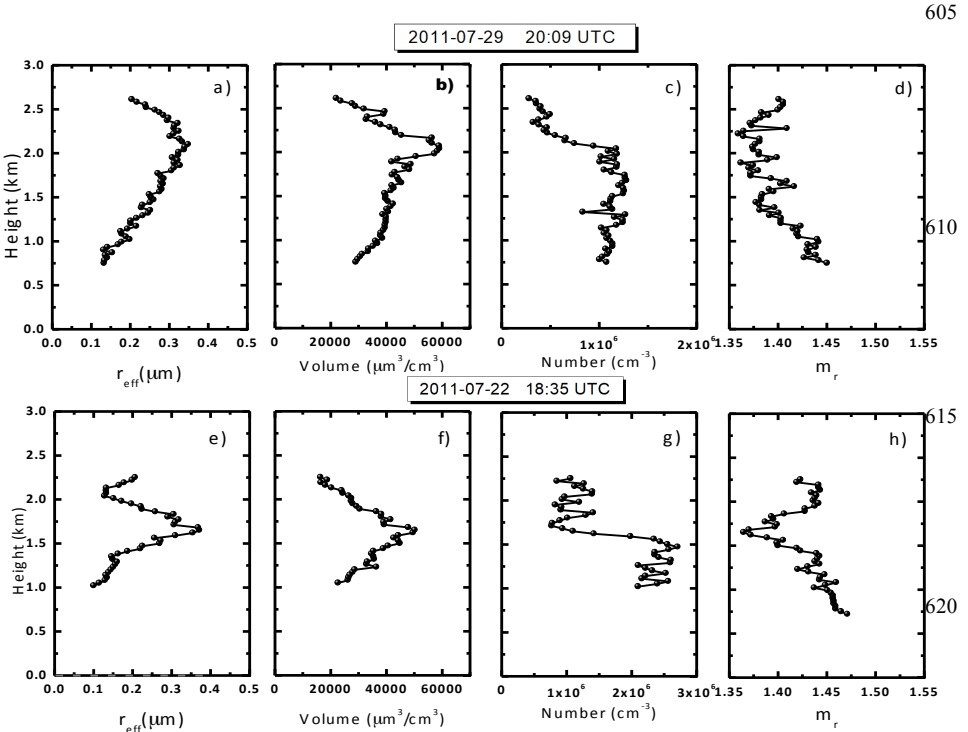

**Figure 8:** Retrieved effective radius ($r_{eff}$), volume and number concentrations and real part of refractive index ($m_r$) from the stand-alone 3β+2α lidar inversion for (a-d) 29th July 2011 and for (e-f) 22nd July 2001.




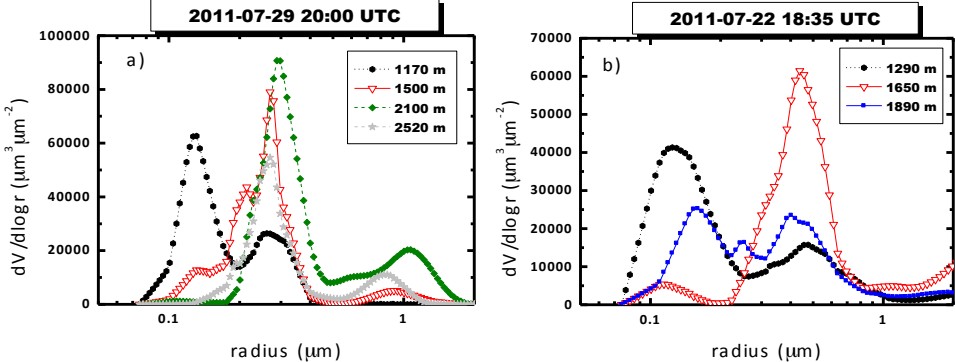

**Figure 9:** Particle volume size distributions at different levels obtained from the stand-alone 3β+2α lidar inversion for (a) 29th July 2011 and for (b) 22nd July 2001.

Referring to Figure 8, on both days $r_{eff}$ shows an increase with altitude from values of ~ 0.12 μm at 1 km altitude to values around 0.3-0.35 μm at the top of the boundary layer when maximum hygroscopic growth was achieved. Above the planetary boundary layer, the decrease of $r_{eff}$ with altitude is clear because above this level relative humidity drops drastically thus eliminating hygroscopic growth. For volume concentration Figure 8 reveals an increase with altitude that is consistent with the increase of extinction with altitude and with the changes in particle volume size distributions. We note that now

volume concentration is for ambient conditions while the results obtained from P-3B measurements were for dry conditions, which explains the differences with altitude. For number concentrations, which is the parameter with the largest error of ~ 100% (Whiteman et al., 2018), a pattern with approximately constant values is observed in the planetary boundary layer in agreement with the expectation of approximately the same number of particles for well-mixed conditions. Finally, for $m_r$ we observe a decrease with altitude from values ~ 1.45 to values close to ~ 1.35 which is typical for hydrated particles. We note

that retrieved $m_i$ was below 0.005 for all cases which produced high SSA (>0.98). The range of retrieved $r_{eff}$, refractive indices and SSA are typical of a mixture of sulfate and water soluble organic carbon (e.g. Chin et al., 2002) which is the predominant chemical composition from Table 4.

       Figure 9 illustrates the displacement of size distribution to large radius with altitudes within the boundary layer, which is the typical pattern expected (e.g. Schafer et al., 2008) and agrees with the increase in radius with altitude observed

in Figure 8 and associated with hygroscopic growth. The size distributions in Figure 9 suggest a second fine mode (particles with radius below 0.5 um) which could be an indication of aerosol-fog modification (e.g. Eck et al., 2012) although such a conclusion must be considered tentative due to uncertainties (~100%) in the retrieved size distribution. Above the planetary boundary layer, the size distributions are clearly displaced toward smaller radii.


### 3.3 Aerosol spatial distribution through the day

Airborne measurements acquired by the HSRL-1 system provided extended records of the evolution of aerosol vertical distribution during the field campaign. For example, Figure 10 shows more than 8 continuous hours of α(532 nm) from HSRL-1 measurements on July 29 and 22. Data shown in Figure 10 are only cloud-free data with white vertical lines illustrating the times when the aircraft flew over NASA GSFC. The large gaps around 19 UTC on July 29 and 17 UTC on July 22 are when the UC-12 airplane landed and refueled between the morning and afternoon flights, while the small gaps

correspond to cloud filtered data.

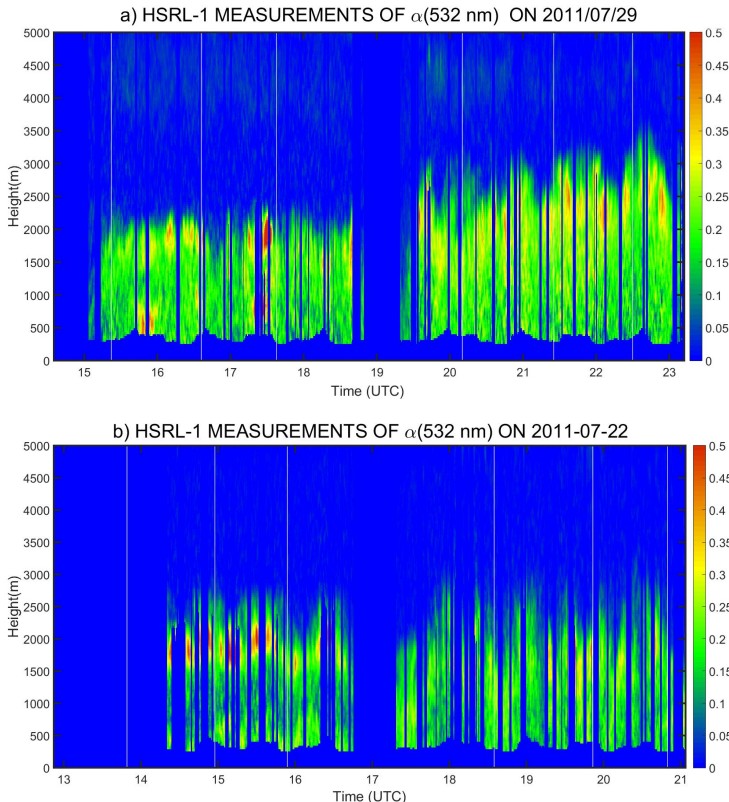

**Figure 10:** Temporal evolution of aerosol extinction coefficient (α) at 532 nm obtained by HSRL-1 system over the Baltimore-Washington area on (a) 29<sup>th</sup> July 2011 and (b) 22<sup>nd</sup> July 2011. Vertical white lines represents times when the system flew over NASA
Goddard Space Flight Center



For July 29 at 18:00 UTC Figure 10a reveals some decoupled aerosol layers within the planetary boundary layer that agrees with the patterns observed in Figure 4. After that time, there are many cloud-filtered data that agree with the convective clouds observed in Figure 4 as well. In particular, though, it should be noted that the aircraft measurements reveal that after approximately 19:00 UTC on both days, but particularly on July 29, there were increases of aerosol extinction with height within the planetary boundary layer. These increases agree again with those observed in Figure 4 and can be associated with aerosol hygroscopic growth. On the other hand, data from Figure 10b complement the sparse data obtained from ground-based lidar measurements on July 22. Again, some decoupled aerosol layers are observed early in the morning. But on that day there were more cloud-affected data. Nevertheless, after 18:00 UTC an increase in α(532 nm) with height is again observed within the planetary boundary layer consistent with the aerosol hygroscopicity previously studied. For much of the HSRL-1 datasets on July 29 and 22 daytime well-mixed conditions associated with convective processes were present with relative humidity increasing with altitude (Figure 3). Also, chemical composition measurements revealed the presence of hygroscopic aerosols (Table 4). Therefore, the similarity in aerosol vertical distribution patterns between both days can be interpreted as indicating the presence of swelling aerosols resulting in an aerosol profile characterized by an increase in backscattering and extinction with height. Based on the aircraft measurements, we take that pattern to be representative of the extended area and not just over the specific ground-based lidar site.

The large number of stations deployed by AERONET-DRAGON permits us to study how aerosol hygroscopicity affected aerosol optical depth (AOD), representative of the entire column, over the study region. Figure 11 shows the color-map of hourly-mean AODs for the Washington-Baltimore region on July 29, 2011. We performed gridded linear interpolations between the stations to obtain the color maps shown (see Fig. 1c for an illustration of the AERONET-DRAGON stations). We note that the number of stations for the interpolations differed among different hours because of the presence of partly cloudy skies that obscured the sun and thus prevented AERONET-DRAGON measurements from being performed. What Figure 11 clearly reveals is an increase in AOD through the day reaching the maximum values in the evening (AOD ~ 0.7-0.8) in spite of some spikes in the northern locations that could be associated with automobile traffic or other local anthropogenic emissions in the region. The evening values of AOD can be as much as twice as those in the early morning. If we assume that atmospheric conditions from Figure 3 can be extrapolated to the entire region, then the well-mixed conditions during the evening can better explain the more regionally uniform AOD values than those obtained during the morning.

The spectral deconvolution algorithm (O'Neill et al., 2003) was used to separate AOD into fine ($AOD_{fine}$) and coarse ($AOD_{coarse}$) mode contributions. Figure 12 displays $AOD_{fine}$ temporal evolution for July 29, 2011 ($AOD_{coarse}$ graphs not shown for brevity). Again, the figures show the values for the entire Washington-Baltimore region and data were gridded using linear interpolations. The similarity of figures 11 and 12 indicate a predominance of $AOD_{fine}$ over $AOD_{coarse}$ during the entire day. Actually, $AOD_{coarse}$ was mostly below 0.06 –with some exceptions in the northern locations in the morning perhaps due to local sources (e.g. road traffic) but not representative of the entire region. The values of $AOD_{fine}$ are





comparable to those of total AOD and were seen to increase during the day, with larger values during the evening (~ 0.7-0.8) than during the morning (~ 0.4-0.6). These patterns reveal that the increase in the AOD during the day is associated mainly with changes in the fine mode. But now, combining the information of aerosol hygroscopicity previously obtained from lidar measurements and from the analyses of aerosol hygroscopicity in section 3.2, we can conclude that changes in the AOD during the day are mainly associated with changes induced by aerosol hygroscopicity that occurred when well-mixed conditions were achieved. We conclude that these changes in the AOD occurred over an extended region. Similar patterns in AOD were obtained for July 22 but the graphs are not shown for brevity.

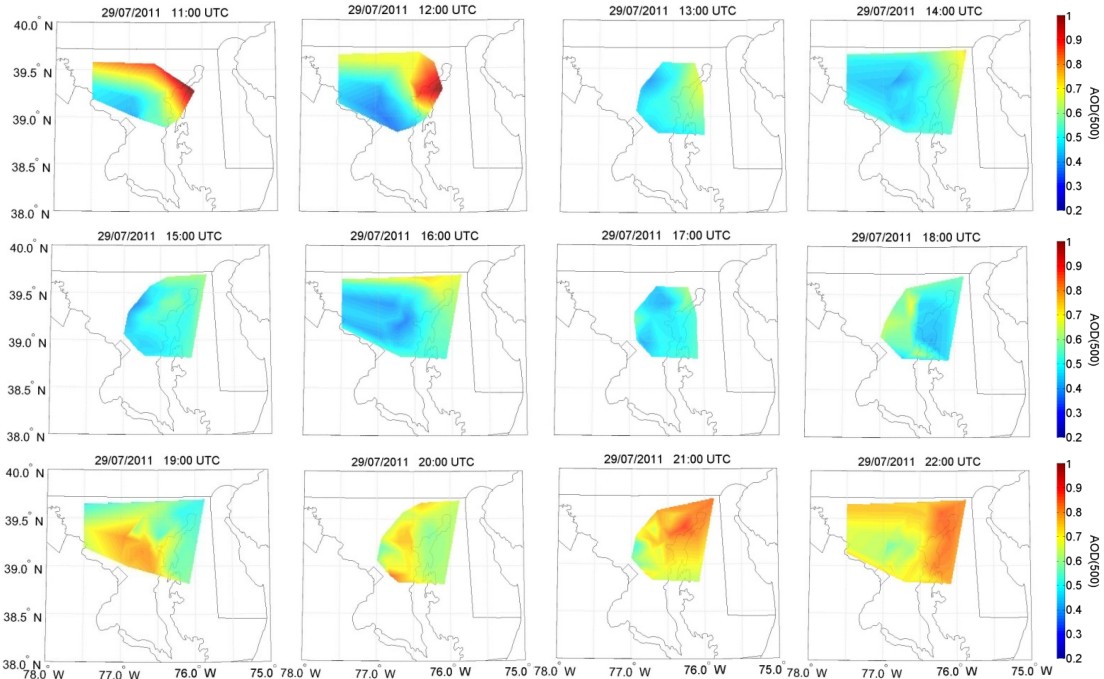

**Figure 11:** Temporal evolution of aerosol optical depth (AOD) at 500 nm from AERONET-DRAGON over the Washington-Baltimore area on 29th July 2011.

Figure 13 shows hourly-mean volume aerosol size distributions from AERONET-DRAGON Level 2.0 data, both for July 29 and 22. Unfortunately, due to partly cloudy skies there were some times during the day when there were no retrievals, particularly on July 22. Nevertheless, the volume size distributions clearly illustrate an increase in the fine mode for both days that is responsible for majority of the total increase in aerosol AOD during the day.






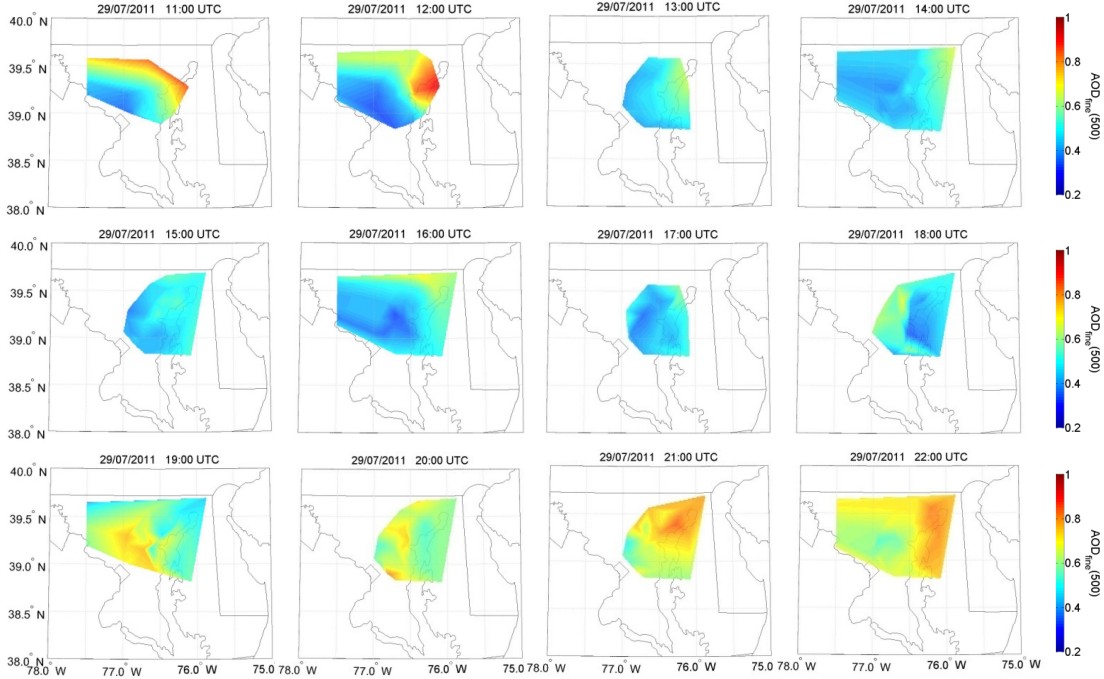

**Figure 12:** Temporal evolution of fine mode aerosol optical depth (AOD$_{fine}$) at 500 nm from AERONET-DRAGON over the Washington-Baltimore area on 29th July 2011





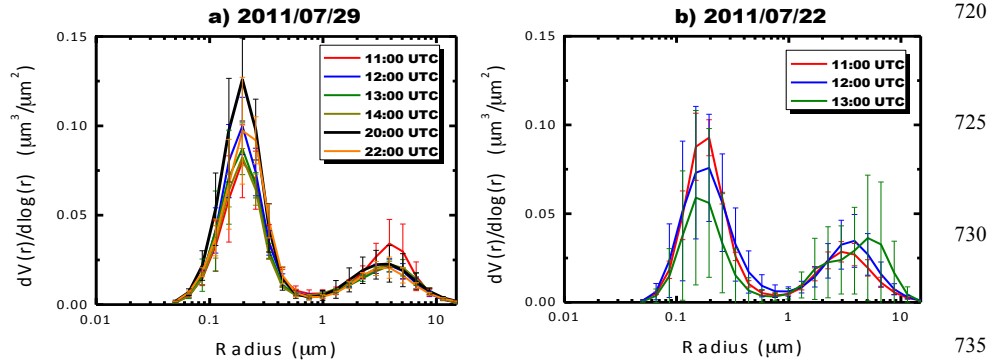

**Figure 13:** Mean hourly values of aerosol size distributions from AERONET-DRAGON stations over the Washington-Baltimore region on (a) 29$^{th}$ July 2011 and (b) 22$^{nd}$ July 2011.





Mean optical and microphysical results obtained from AERONET Level 2.0 inversions are summarized in Table 5 differentiating between morning (before 13:00 UTC) and evening (after 19:00 UTC) values. Also, results in Table 5 are shown for 532 nm using linear interpolations between retrieved values at 440 and 670 nm, since no significant wavelength-dependence was found. For $m_r$ the retrieved values are between 1.36 and 1.39 and are typical of highly hydrated particles. The very low values of $m_i$ ($\leq$0.003) and high values of SSA ($\geq$0.98) indicate that absorption is negligible as well. All of these values of refractive index and SSA are typical for highly hydrated particles (e.g. Chin et al., 2002). For the microphysical

properties, the effective radius is also relatively high although within the fine mode ($r_{eff} \sim$ 0.23-0.25 μm). Values of $r_{fine}$ are similarly high ($\sim$ 0.17 μm). But these ranges of $r_{eff}$ and $r_{fine}$ are typical of a size distribution of highly hydrated particles (e.g. Dubovik et al., 2002; Perez-Ramirez et al., 2017). The predominance of $V_{fine}$ over $V_{coarse}$ is clear implying a negligible coarse mode.. But AERONET retrievals do not show significant differences between morning and evening values and do not refer significant differences associated with the presence or lack of hygroscopic growth as revealed by Fig. 11 and Fig. 13. The

lack of measurements in the afternoon could explain that as revealed in Fig. 12 when maximum volume in fine mode is found during retrievals close to noon. The large number of stations together with the errors associated with AERONET inversions can mask temporal changes with relative humidity. Another point is that AERONET retrievals are effective values representative of the entire atmospheric column, which can mask specific variations in aerosol microphysical properties with altitude. Nevertheless, it should be noted that AERONET retrieved parameters and those from stand-alone

3β+2α lidar inversion shown in Figure 8 are consistent with each other and representative of aerosol hygroscopic growth. Therefore, the combined lidar and AERONET retrievals presented serve to further understand changes in aerosol microphysical properties with relative humidity.

| | 2011/07/29 | | 2011/07/22 | |
|---|---|---|---|---|
| | **Morning** | **Evening** | **Morning** | **Evening** |
| $m_r$ (500 nm) | 1.38 ± 0.03 | 1.39 ± 0.03 | 1.36 ± 0.02 | NA |
| $m_i$ (500 nm) | 0.003± 0.002 | 0.003± 0.002 | 0.003± 0.001 | NA |
| SSA (500 nm) | 0.98 ± 0.02 | 0.98 ± 0.02 | 0.98 ± 0.02 | NA |
| g (500 nm) | 0.68 ± 0.02 | 0.68 ± 0.02 | 0.69 ± 0.01 | NA |
| $r_{eff}$(μm) | 0.23 ± 0.02 | 0.22 ± 0.03 | 0.26 ± 0.03 | NA |
| V (μm³/μm²) | 0.14 ± 0.04 | 0.13 ± 0.03 | 0.14 ± 0.07 | NA |
| $r_{fine}$(μm) | 0.18 ± 0.01 | 0.17 ± 0.01 | 0.17 ± 0.01 | NA |
| $V_{fine}$(μm³/μm²) | 0.12 ± 0.03 | 0.09 ± 0.05 | 0.10 ± 0.05 | NA |
| $r_{coarse}$(μm) | 2.7 ± 0.3 | 2.8 ± 0.2 | 2.9 ± 0.3 | NA |
| $V_{coarse}$(μm³/μm²) | 0.03 ± 0.01 | 0.04 ± 0.02 | 0.05 ± 0.02 | NA |

**Table 5:** Mean real refractive index ($m_r$), imaginary refractive index ($m_i$), single scattering albedo (SSA), asymmetry factor (g), effective radius ($r_{eff}$), particle volume (V), fine mode radius ($r_{fine}$) and volume ($V_{fine}$) and coarse mode radius ($r_{coarse}$) and volume ($V_{coarse}$) obtained
from AERONET Level 2.0 inversions on 29[th] July 2011 and on 22[nd] July 2011.

The analyses of the in-situ measurements taken from the P-3B aid the understanding of aerosol hygroscopic effects. Figure 14 shows mean hourly values of $\alpha_{amb}$(532 nm), γ and f(RH) for both July 29 and 22. We note that the values of $\alpha_{amb}$(532 nm) are computed by adding the absorption coefficient to $\sigma_{amb}$(532 nm) which is obtained using Eq. 1 from the



measured $\sigma_{dry}$(532 nm), γ and relative humidity. Data are presented for three different altitudes: near surface (altitude below 1 km), in the planetary boundary layer (1-2 km for July 29 and 1-1.8 km for July 22) and above the planetary boundary layer (>2.0 km for July 29 July and > 1.8 km for July 22). The analyses of $\alpha_{amb}$(532 nm) clearly reveals the importance of hygroscopic growth as ambient values are generally 1.8 times the dry values, which is consistent with the measured f(RH). Generally the larger values of $\alpha_{amb}$(532 nm) are obtained in the altitude range considered to be in the planetary boundary

layer, which is consistent with the previous finding of larger extinction in that region due to hygroscopic growth for the atmospheric conditions present on both days (Figure 3). Also, both days show that below the top of the boundary layer, $\alpha_{amb}$(532 nm) is approximately constant for both dry and humid conditions, but above the boundary layer there is an increase with time after 20:00 UTC for July 29 – the lack of data for these times on July 22 may suppress that pattern. Actually, the fact that the last measurement at 22:00 UTC above the boundary layer has larger extinction values than the rest can be

explained by the increase of PBL height (see Fig 10a) and thus the P3B airplane is sampling the swollen aerosols that earlier in the day were not sampled.

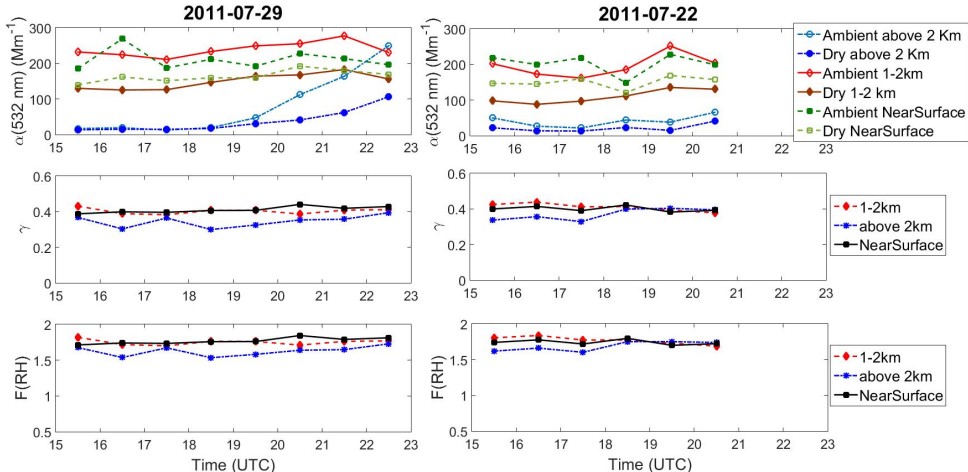

**Figure 14:** Hourly mean values of aerosol extinction at 532 nm (α(532 nm)) and hygroscopicity parameters γ and F(RH) for both 29[th] and 22[nd] July. Data are presented for three different altitudes: Near surface (altitude below 1 km), in planetary boundary layer (1-2 Km for 29[th] July and 1-1.8 Km for 22[nd] July) and above the planetary boundary layer (>2.0 Km for 29[th] July and > 1.8 Km for 22[nd] July).


      Another important result from Figure 14 is that during the entire day there are relatively constant values of γ (~ 0.35-0.41) and f(RH) (~ 1.6-1.8) for both days. This implies that hygroscopic aerosols were present during the entire measurement periods and over the broad study region, and the combination of the well-mixed atmospheric conditions layers

and increasing relative humidity with height created the conditions for aerosol swelling and the increase in aerosol backscattering and extinction with height. These conditions are found typically in the afternoon/evening and are associated with convective conditions. Early in the morning, the hygroscopic growth can also be present but because of the lack of well-



mixed conditions the effect on atmospheric extinction and backscattering cannot be predicted using the procedure here and depends on the characteristics of each day.

Figure 15 shows the hourly mean number of particles obtained from the P-3B for the same vertical intervals as in Figure 14. Both UHSAS (radius between 0.06 – 1µm) and LAS (radius between 0.9 – 7.7µm) measure about the same number of particles and the patterns of the values from both instruments are very similar near the surface and in the planetary boundary layer with stable values through the entire measurement period. These results are consistent with those from Figure 8 and imply that a similar concentration of particles was present in the atmosphere during the day and that changes in aerosol

extinction, backscattering and AOD are explained by aerosol hygroscopicity. Above the planetary boundary layer the number of particles drops drastically and, as seen in Figure 3, the relative humidity drops significantly. Finally, it is important to note that the SMPS always indicates a larger number of particles than for UHSAS and LAS which shows significant variability during the day, particularly for July 29. But the SMPS is representative of ultrafine particles (radius between 10 – 300 nm) that are not detected by lidar measurements because of the lack of counting efficiency at the emission

lidar wavelengths and comparisons are not straightforward.

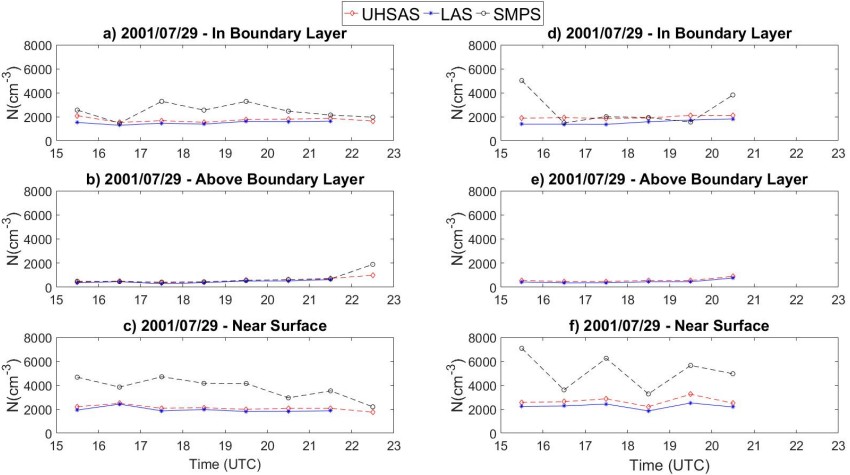

**Figure 15:** Hourly mean values of mass of the different species that form aerosol particles measured by the P-3B airplane on 29[th] and 22[nd] July 2011. Data are presented for three different altitudes: Near surface (altitude below 1 km), in planetary boundary layer (1-2 Km for 29[th] July and 1-1.8 Km for 22[nd] July) and above the planetary boundary layer (>2.0 Km for 29[th] July and > 1.8 Km for 22[nd] July).


The evolution of chemical composition of aerosol particles serves to better understand aerosol hygroscopicity for the two study days. Figure 16 shows hourly means of the main species (mass above 0.03 µg/m³). As in the previous Figures data are shown for near surface, in the planetary boundary layer and above the planetary boundary layer. For any day and layer sulfate is by far the predominant specie followed by water soluble organic carbon and ammonium. As we saw in Table



4, these species are highly hygroscopic and explain the aerosol hygroscopicity previously observed. The total mass is stable throughout the day except for a slight increase in sulfate total mass after 18:00 UTC which can be explained as an accumulation of sulfate particulate due anthropogenic emissions during the day (e.g. Fitzgerald et al., 1982). Ammonium also shows a very slight increase late in the evening for July 29. We note that standard deviations of mean values were below 10% confirming the stability in mass amount and percentage and implying that the results are representative of the entire

study region. Therefore, assuming that sulfate, ammonium and water vapor organic carbon emissions are of anthropogenic origin we can conclude that the impact of these emissions on aerosol backscattering and extinction profiles and eventually on AOD is enhanced by aerosol hygroscopicity.

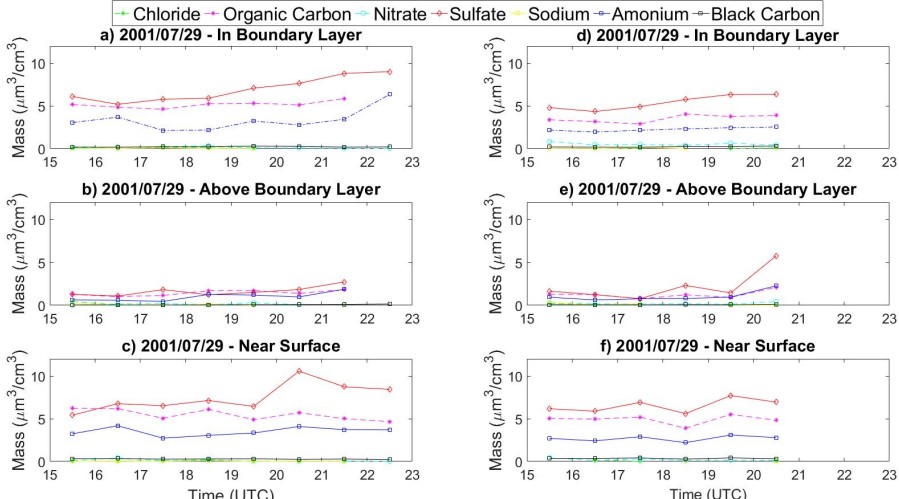

**Figure 16:** Hourly mean values total number of particles obtained by SMPS, UHSAS and LAS instruments onboard the P-3B airplane and
acquired on 29[th] and 22[nd] July 2011. Data are presented for three different altitudes: Near surface (altitude below 1 km), in planetary boundary layer (1-2 Km for 29[th] July and 1-1.8 Km for 22[nd] July) and above the planetary boundary layer (>2.0 Km for 29[th] July and > 1.8 Km for 22[nd] July).

### 3.4 Aerosol radiative impact

Figure 17 shows aerosol radiative effect (ARE) profiles while Figure 18 shows the heating rate (HR) profiles, for
both July 22 and 29, 2011. In the figures are represented ARE both differentiating among the wavelengths of lidar measurements (355, 532 and 1064 nm) and the integrated values for the shortwave region (280-3000 nm). The computations were done using LibRadtram following the configuration described in section 2.4 and using the 3β+2α vertical profiles for the hygroscopic growth cases illustrated in Figure 5. Inputs of temperature and humidity profiles were those obtained from radiosonde measurements launched very close in time with lidar measurements (Figure 3). Extinction and backscattering
measurements at ambient conditions were used in the computations. But the equivalent backscattering profiles for 'dry'





conditions were also computed using the Hänel equations (Eq. 5) and the corresponding 'γ(λ)' obtained in Figure 6 and RH measurements, where the reference values $\beta_{ref}$ and $RH_{ref}$ were those obtained at the lowest altitude. For the extinction profiles at dry conditions, we used again the Hänel equation but replaced backscattering by extinction, and assumed the same hygroscopic growth factor 'γ(λ)'. For the incomplete overlap region (approximately the first kilometer), α and β were

computed using the Hänel equation with the hygroscopicity parameters 'γ(λ)' and the measured RH in this first kilometer. Below 3 km where most of the aerosol was found, for July 22 the obtained aerosol optical depths at 355 and 532 nm were approximately 0.50 and 0.36 for ambient conditions and of 0.33 and 0.27 for dry conditions, while for July 29 the AODs were approximately 0.95 and 0.57 for ambient conditions and 0.76 and 0.4 for dry conditions. Solar zenith angles were 23.24° for July 22 and of 42.69° for July 29. Because we are using the same methodology for ARE computations, possible

differences in ARE computations between ambient and dry conditions for the same profile should be independent of the errors associated with the methodology for ARE computations (e.g. Sicard et al., 2014; Granados-Muñoz et al., 2019).

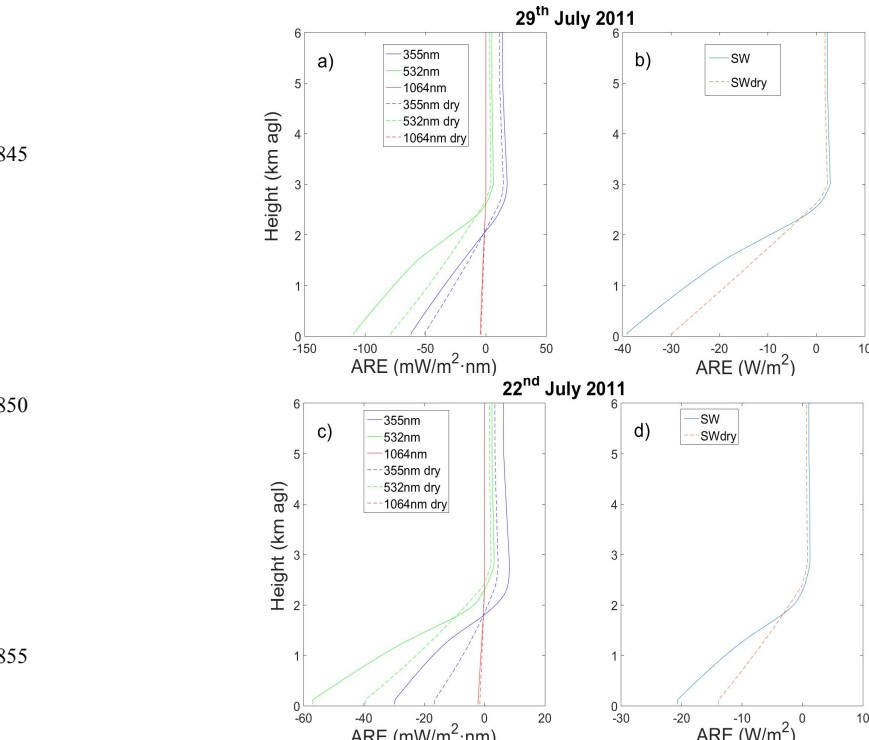




**Figure 17:** Vertical profiles of aerosol radiative effect (ARE) from the lidar measurements affected by aerosol hygroscopic growth. Data are presented (left) for the measured lidar measurements (355, 532 and 1064 nm) and (right) integrated for the shortwave range (280 – 860    3000 nm).

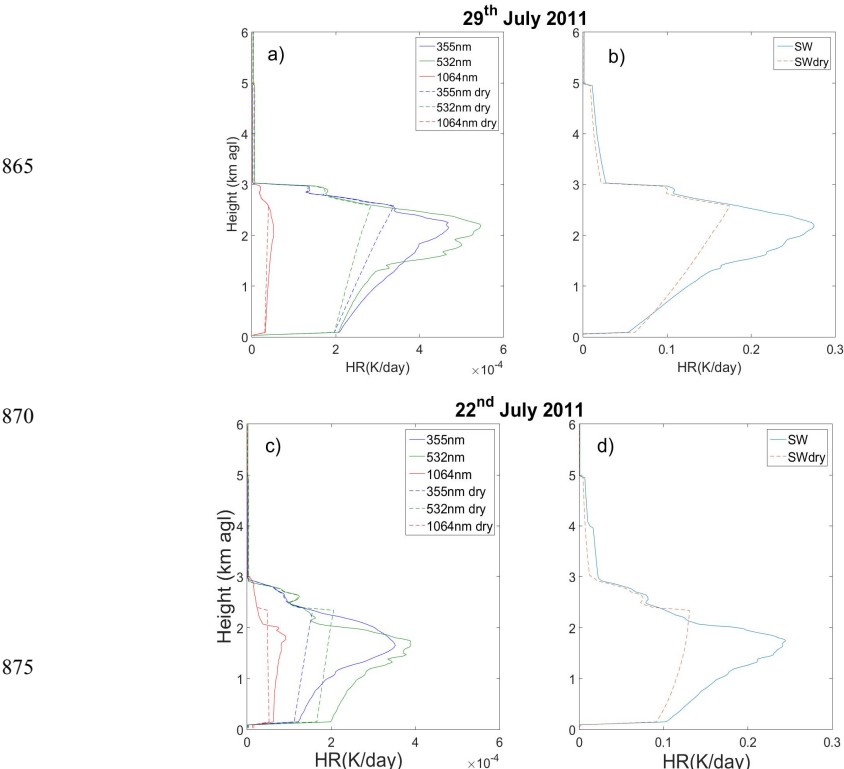




**Figure 18:** Vertical profiles of aerosol heating rates from the lidar measurements affected by aerosol hygroscopic growth. Data are presented for the measured lidar measurements (355, 532 and 1054 nm) and integrated for the shortwave range (280 – 3000 nm).


Figure 17 clearly demonstrates a wavelength dependence of ARE for both days, with the largest cooling effect obtained at 532 nm on both days followed by 355 nm. Actually, systematic lower values of ARE are observed at ambient conditions below the top of PBL. At the surface the differences of ambient minus dry ARE are the largest and take values of -17.4 and -13.2 W/m$^2$/nm at 532 and 355 nm for July 29 and of -30.85 and -11.2W/m$^2$/nm at 532 and 355 nm for July 29

July. The systematic lower values of ARE (in absolute value) for July 29 are explained by the larger aerosol load and solar zenith angle for that day. Above the PBL there are no important differences between ambient and dry profiles for any wavelength (values of ~ 2.5W/m$^2$/nm at 532 nm and ~ 3.5 W/m$^2$/nm at 355 nm), with differences between both days negligible. Values at 1064 nm are basically zero for both days both for ambient and dry conditions independently of altitude, and therefore we can conclude that ARE for these types of particles (mixture of sulfates and water soluble organic carbon) is





only sensitive in the visible and UV region. This has to be taken into account when analyzing the SW profiles that reported an important difference between ambient and dry profiles where the maximum difference is at the surface (approximately of -10 and -7 W/m$^2$ on July 29 and 22, respectively) and minimum close to zero near the top of the PBL. Actually, above these altitudes small constant values are observed with height with no differences between dry and ambient conditions. Therefore, we conclude that aerosol hygroscopic growth is the cause of the larger cooling effect for the types of particles analyzed here.

Comparisons of ARE with other aerosol types are not straightforward because of the dependences with AOD and solar zenith angle, although the reported cooling effects agree with other studies (e.g. Di Biagio et al., 2009; Huang et al., 2009; Bhawar et al., 2016; Mallet et al., 2016; Wang et al., 2020)

       Figure 18 also shows a wavelength dependence in HRs below the PBL with positive values above $2 \cdot 10^{-4}$ K/day for 355 and 532 nm, with HRs very close to zero at 1064 nm. For the aerosol-free region above PBL the computed HRs are

basically zero for all wavelengths. Differences between ambient and dry conditions are found for 355 and 532 nm in the region close to the top of the PBL where there was more hygroscopic growth. These differences between ambient and dry conditions are approximately $2.11 \cdot 10^{-4}$ and $1.98 \cdot 10^{-4}$ K/day at 532 and 355 nm for July 29 and of $2.78 \cdot 10^{-4}$ and $1.51 \cdot 10^{-4}$ K/day at 532 and 355 nm for July 29. The integrated SW values show similar patterns on both days with an increase from minimum values at the surface (~ 0.05 K/day) to maximum values near the top of the PBL where the differences between

ambient and dry conditions are maximized (~0.12 K/day) and associated again with aerosol hygroscopic growth. These positive HRs for the SW region agree with other studies (Mallet et al., 2008; Lemaitre et al., 2010; Perrone et al., 2012; Meloni et al., 2015; Granados-Muñoz et al., 2019) although the aerosol types in this study are different.

## 5 Summary and conclusions

       This work has focused on the study of aerosol hygroscopic properties during the DISCOVER-AQ 2011 field

campaign in the Washington D.C. – Baltimore metropolitan area. During the campaign a unique dataset was available: The NASA P-3B airplane deployed in-situ instrumentation that permitted the characterization of aerosol hygroscopic growth and other aerosol optical, microphysical and chemical properties, while the HSRL-1 lidar system was deployed on the NASA UC-12 airplane for continuous characterization of backscattering (β) and extinction (α) profiles at 532 nm. Ground-based measurements included the multiwavelength Raman lidar at NASA Goddard Space Flight Center (GSFC) that acquired

independent profiles of aerosol extinction and backscattering at 355 nm. Both Raman and HSRL-1 system obtained aerosol backscattering at 1064 nm. Combined Raman and HSRL-1 measurements made during UC-12 overpasses at GSFC provided the 3β+2α configuration used for the retrieval of vertically-resolved aerosol microphysical properties, and the study presented here serve to complement the hybrid configuration of lidars presented by Sawamura et al., (2014) over the same region The deployment of AERONET-DRAGON with more than 40 instruments in the region permitted a continuous

characterization of columnar aerosol properties and the study of their spatial representativeness and variability.



For the study of aerosol hygroscopic growth with lidar measurements, stable and well-mixed conditions were needed. Radiosondes launched from the Howard University Beltsville Campus (HUBC) permitted the characterization of the state of temperature and humidity in the atmosphere every four hours, and cases that fulfill the previous conditions and with available lidar measurements were identified for July 22 and 29, 2011. The focus of the study was on July 29 because

multiwavelength Raman lidar measurements were available for approximately 10 daytime hours. On 22 July only 1.5 hour of Raman measurements were available and thus only serve to complement the results of the other day. On both days air masses had their origin close to the Ohio river valley and there was no significant transport of aerosol at altitudes above the planetary boundary layer (PBL). Mean daily aerosol optical depths (AOD) at 500 nm were approximately 0.52 for July 29 and 0.42 for July 22, with both days having an Angström exponent of ~1.8. The thermodynamic state of the atmosphere was

characterized by well-mixed conditions in the afternoon below the top of the planetary boundary layer, with stable water vapour mixing ratio and relative humidity increasing from the surface (30-40 %) up to the top of the planetary boundary layer (values between 80-90%).

For the study of aerosol hygroscopicity the Hänel equation $f(RH) = ((1-RH)/(1-RH_{ref}))^{-\gamma}$ was used, with $f(RH)$ being the ratio between humid ($\beta_{amb}$) and the reference dry ($\beta_{ref}$) backscattering coefficients, RH and $RH_{ref}$ the relative humidity both at ambient and reference conditions, and $\gamma$ the hygroscopicity factor that depends on the type of particles. From lidar

measurements, linear fits were performed taking logarithms of both sides of the Hänel equation to obtain $\gamma$ where the lowest height with measured backscattering was taken as the reference to compute $\beta_{ref}$ and $RH_{ref}$.. Computations of $\gamma$ revealed a wavelength-dependence for 355, 532 and 1064 nm of 0.46, 0.39 and 0.31 on July 29 and of 0.65, 0.38 and 0.37 on July 22, although close to the uncertainty of the method that was demonstrated to be approximately 15% through different

simulations. These values of $\gamma$ were confirmed by correlative spirals at GSFC by the P-3B using a pair of nephelometers that provided mean values of ~ 0.41 at 532 nm, with the lidar retrieved values of $\gamma$ agreeing within the estimated uncertainties of 15%. Therefore, these results serve as validation of measurements of aerosol hygroscopicity from lidar measurements and confirm that the increase of aerosol backscattering with height measured by lidar measurements was due to aerosol hygroscopic growth effects. The extinction coefficients were observed to increase with altitude which can be also explained

by the effects of aerosol hygroscopicity.

The $3\beta+2\alpha$ lidar measurements obtained by the combination of ground based Mie-Raman and airborne HSRL-1 lidar systems during periods of aerosol hygroscopic growth were used as inputs to the regularization technique to retrieve aerosol microphysical properties. Effective radius ($r_{eff}$) showed an increase with altitude from dry values around 0.10-0.15 μm to humidified values of 0.30-0.4 μm. Volume concentration showed an increase with height similar to that of the

extinction coefficients. Number concentration remained constant which agrees with the hypothesis of the same number of particles under well-mixed conditions, and the same was found by P-3B measurements. The real refractive index ($m_r$) showed a decrease with height from dry values of 1.45-1.50 to humidified values near to 1.35 which are the typical for highly hydrated particles. Single scattering albedo (SSA) both for dry and humidified conditions was above 0.99 which



indicates essentially no absorption. The complementary chemical analyses by P-3B indicated that the predominant species

were sulfate and water vapor soluble organic carbon which are known to be highly hygroscopic and non-absorbing. Therefore, the combination of 3β+2α lidar retrievals with airborne measurements provided a unique closure study for aerosol hygroscopic growth characterization.

The analyses of AERONET-DRAGON data reported an increase of AOD throughout the day particularly for the fine mode. This was observed for the entire Washington D.C. – Baltimore metropolitan area. In spite of partly cloudy skies,

the large number of instruments deployed provided sufficient AERONET inversions to obtain mean $m_r$ ~ 1.38, $r_{eff}$ ~ 0.24 μm and SSA ~ 0.98 that are consistent with these obtained by lidar retrievals and confirmed aerosol hygroscopic growth over the region. On the other hand, P-3B measurements during the entire day reveal the same Hänel hygroscopicity parameters with time and altitude, which confirms that aerosol hygroscopicity was present during the entire day. The P-3B measurements also indicated a constant number of particles and concentration of main species (sulfate and water soluble

organic carbon) during the day. Thus, combining all AERONET-DRAGON, P-3B and lidar measurements we can conclude that the increase of AOD during the day and the increase of aerosol backscattering and extinction with height were directly associated with aerosol hygroscopic growth.

Through the use of the Hänel equations, assuming the same hygroscopic growth parameters γ(λ) for extinction and backscattering and with the relative humidity profiles it was possible to compute extinction and backscattering coefficients

for both ambient and dry conditions. This permitted the isolation of the effect of aerosol hygroscopicity and the study of their impacts on aerosol radiative effects (ARE) and heating rates (HR), which were computed using the libRadtram radiative transfer code. Computations revealed a cooler surface for the shortwave range (differences between 7-10 W/m$^2$ depending on aerosol load) and a larger heating near at the top of the PBL (differences of approximately 0.12 K/day) directly associated with aerosol hygroscopic growth. Our results can be extrapolated to other areas where aerosol particles have

similar chemical compositions, but further studies are needed for other types of particles such of absorbing particles. Studying how the presence of a large amount of aerosol hygroscopic particles affects the development of convective clouds is a challenge for future research.

**Acknowledgements**

This work was supported by the Marie Skłodowska-Curie Research Innovation and Staff Exchange (RISE) GRASP-ACE

(grant agreement No 778349) and by the Spanish Ministry of Economy and Competitiveness (RTI2018.101154.A.I00). The authors thankfully acknowledge the AERONET team for maintaining the stations used in this work and to the NOAA Air Research Laboratory for providing the HYSPLIT model. We are grateful for the HSRL and LARGE teams and the NASA Langley King Air crew for the support of HSRL measurements and operations during the NASA DISCOVER-AQ field misión. The HSRL-1 and LARGE DISCOVER-AQ data used here are publicly available at the NASA Archive https://www-





air.larc.nasa.gov/cgi-bin/ArcView/discover-aq.dc-2011. We are grateful for the efforts of the dedicated team at the Howard
       University Beltsville campus for the radiosonde launches that supported so much of this study.

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
