# Peer review of "Spatiotemporal changes in aerosol properties by hygroscopic growth and impacts on radiative forcing and heating rates during DISCOVER-AQ 2011."

_Atmospheric Chemistry and Physics, 2021_

## Author Comment (AC1)

**The authors greatly acknowledge the anonymous reviewer for carefully reading the manuscript and providing constructive comments. This document contains the authors' responses to comments.**

**General Remarks:**

Aerosol hygroscopicity, which describes water uptake by particles in a humid atmosphere, is a crucial feature that determines the physical properties of particles and subsequently their roles in atmospheric radiation. By integrating a series of lidar, in-situ, and radiosonde measurements from DISCOVER-AQ Washington aircraft campaign, GSFC and AERONET-DRAGON ground stations, and launched radiosondes in site, this study revealed aerosol hygroscopicity vertically in two selected humid days during DISCOVER-AQ Washington campaign. The study explained the significant increase of extinction and backscattering with height owing to aerosol hygroscopicity. The study also revealed that aerosol hygroscopicity is responsible for larger cooling effects in the shortwave range near the ground, and a larger warming near the top of PBL where aerosol hygroscopic growth was highest by using vertically-resolved aerosol extinction/backscattering measured by lidars. This is an interesting and valuable study. The paper is well written. I recommend publishing it in ACP after the authors make the following minor modifications.

**We thank the referee for the positive feedback. In the following lines we address each referee specific comments**

**Specific comments:**

1. Abstract line 24: Please explain radiosonde measurement before giving its application.

**That line will be modify in the revised manuscript to point out that lidar measurements served to retrieve aerosol extinction and backscattering profiles while radiosondes provided measurements of relative humidity and temperature**

2. Abstract lines 29-30: Need more discussion for this conclusion, see details in specific comment 15

**We clarified in the abstract section that aerosol hygroscopicity pattern was identified as a possible explanation for aerosol optical depth increase during the day, particularly for fine mode**

3. Page 3 line 3: A good reference you may want to cite here:

Myhre, G., B. H., Samset, M. Schulz, Y. Balkanski, S. Bauer, T. K. Berntsen, H. Bian, N. Bellouin, M. Chin, T. Diehl, R. C. Easter, J. Feichter, S. J. Ghan, D. Hauglustaine, T. Iversen, S. Kinne, A. Kirkevåg, J.-F. Lamarque, G. Lin, X. Liu, G. Luo, X. Ma, J. E. Penner, P. J. Rasch, Ø. Seland, R. B. Skeie, P. Stier, T. Takemura, K. Tsigaridis, Z. Wang, L. Xu, H. Yu, F. Yu, J.-H. Yoon, K. Zhang, H. Zhang, and C. Zhou, Radiative forcing of the direct aerosol effect from AeroCom Phase II simulations, Atmos. Chem. Phys., 13, 1853-1877, doi:10.5194/acp-13-1853-2013, 2013.

**That reference will be added to the revised manuscript**

4. Pages 4-6: Could you summarize all the relevant measurements in a table?

**We appreciate referee suggestion but we believe that there are already plenty of graphs and tables in the manuscript and we decided not to add more tables.**

5. Page 6 line 191: Please also give the full name of "Nd:YAG".

**Nd:YAG is a neodymium-doped yttrium aluminum garnet - $Nd:Y_3Al_5O_{12}$ laser. We will introduce this definition in the revised manuscript.**

6. Page 7 line 226 to page 8 line 2: Why not use Raman Lidar measured backscattering at 532 nm?

**For the Raman lidar system, during daytime did not operate the channel at 607 nm (too noisy) that allowed a direct computation of both backscattering and extinction. Indeed, a lidar ratio needs to be assumed for obtaining backscattering 532 nm with that system. However, HSRL-1 did provide direct measurement of backscattering at 532 nm and therefore is assumed as the reference that is used in the inversion by regularization. Clarifications will be added to the revised manuscript.**

7. Page 8 lines 255-256: Please explain briefly how to derive aerosol single scattering albedo and asymmetry factor from the P-3B aircraft and the AERONET inversions.

**Single scattering albedo and asymmetry factor is computed by P-3B from in-situ measurements of extinction and absorption coefficients and particle size distribution. However, AERONET derived values of these parameters are obtained by solving an ill-posed problem that uses direct and sky radiances measurements as inputs. We will introduce such clarifications in the text.**

8. Page 9 lines 266-272: In general, the study of aerosol hygroscopicity does not depend on the atmospheric condition of well-mixed layers. This condition may be important specifically when using Lidar to study aerosol hygroscopicity. It is better to add "Lidar" somewhere in this sentence. Please also clarify the definition of well-mixed layers: the same type of aerosol with altitude, or the constant water vapor mixing ratio with altitude, or both?

**We agree with the referee that well-mixed layers are only required to derive aerosol hygroscopicity with lidar technique. Such well-mixed conditions required both constant water vapor mixing ration and the same type of aerosol with altitude. We will accordingly modify the manuscript to clarify this point**

9. Page 10 lines 1-2: How do you calculate a new γ'? Do you keep previous calculated f(RH) unchanged? How do you map RH in the Gaussian distribution in the calculation? A more detailed explanation for γ' calculated is desired.

**First we start with a fixed value of γ and a set of relative humidities (RH) varying from 20% to 100% that is used to generate f(RH) with Eq. 5. Later, we**

**add errors to RH and obtain a new set of RH'. The new set RH' plus the previous computed f(RH) are used to obtain a new value γ' using the linear fit proposed in Eq. 6. We will accordingly modify the text to add this clarification.**

10. Page 10 line 293: The relative differences in γ should "increase", not "decrease", although the absolute γ value "decreases". Also "relative uncertainties" in the sentence should be changed to "random uncertainties".

**We thank the referee for remarking this confusion in the text. We will add clarifications in the revised manuscript.**

11. Page 10 line 294: Could you elaborate on this sentence? Could random uncertainties be negative? If yes, why do you discuss positive random uncertainties only? Is that because the negative random uncertainties give a symmetric result of its positive counterpart?

**What we summarize in Figure 2a are absolute values of random uncertainties. As referee suggests, the graph will be a symmetric result if we differentiate between positive and negative values. We will again add clarifications in the text.**

12. Page 11 line 322-324: I'm confused by the words "maximum" and "minimum" describing relative differences in γ. The minimum bias should be zero. It may be better to use the range of biases for the discussion.

**Referee is again right. We apologize for the inconsistencies in the text and we will correct the revised manuscript.**

13. Page 13 line 382: The approximate time should be around 17:20 UTC.

**We thank the referee for detecting this typo and we will accordingly correct the manuscript.**

14. Page 19 lines 591-592: What is the measurement evidence to support the conclusion of "with water soluble organic carbon accounting for at least 93% of the total carbon mass"?

**We estimate total carbon mass as the sum of black carbon mass plus water soluble organic carbon. That explains the average 93% contribution of water soluble organic carbon to total carbon accounting all the spirals given in Table 4. Clarifications will be added in the revised manuscript**

15. Page 24 lines 702-205 and Figure 11 & 12: This conclusion is not straightforward based on the hourly AOD change shown in Figure 11 &12 as well as previous discussed aerosol hygroscopicity. The diurnal variation of aerosol dry mass and RH may also contribute to the AOD hourly change. For example, the dry sulfuric aerosol masses shown in Figure 16 start to increase around 19 UTC in both boundary layer and surface. This temporal aerosol mass enhancement also supports the AOD hourly variation. To derive the conclusion more strictly, I suggest rearranging the analysis by first discussing the specific

aerosol properties (i.e., total number of particles, aerosol masses, aerosol extinction and hygroscopicity, etc). I also suggest adding the corresponding discussion for P-3B measured RH here.

**We thank referee suggestions to enrich the discussion. We have moved hourly AOD (Figures 11 and 12) analyses after total number, mass, extinction and hygroscopicity discussions. We have also added an explanation about P-3B measured RH (no significant differences with RH estimates by radiosondes were found)**

16. Page 26 line 768-769: Did P-3B measure aerosol absorption coefficient as well? It would be better to discuss aerosol masses (e.g., Figure 16) first to help understanding the performance of aerosol extinction in Figure 14 such as the small change between the $\alpha_{dry}$ and $\alpha_{amb}$ above 2km even though fRH is pretty high. Does P-3B have RH measurement? If yes, could you demonstrate RH values in the three layers in the figure?

**Single scattering albedo (SSA) was measured by in-situ instrumentation onboard P-3B and for these particular days SSA measured values were above 0.98. For that reason we did not showed SSA in our analyses in Figure 14. But to clarify that absorption is negligible we will specifically note that point in the revised manuscript. On the other hand, P-3B had measurements of relative humidity, but their values agreed with these obtained by the radiosondes measurements (differences were within the uncertainties associated with each methodology). We will modify the text accordingly to clarify that point.**

17. Figures 15 & 16: Please switch the captions of these two figures. Also, the titles should be 2011/07/29 for a,b,c, and 2011/07/22 for d, e, f.

**We thank the referee for detecting this typo.**

18. Page 31 lines 883-887: The unit of the differences of ambient minus dry ARE should be mW/m$^2$/nm, not W/m$^2$/nm, at specific wavelengths (i.e., 532 nm and 355 nm).

**We again thank the referee for detecting this typo.**

19. Page 32 lines 895-896: The impact of aerosol on solar radiative effect depends on not only AOD, but also single scattering albedo (i.e., particle absorption) and asymmetric factor (i.e., particle size and shape). Aerosol particle hygroscopicity impact on all these parameters.

**Referee is right. We will modify the text to avoid any confusion**

20. Page 32 lines 900: Please change "Differences between …" to "The maximum differences between …".

**We again thank the referee suggestion and we correct the manuscript accordingly.**

21. Page 33 lines 942-945: Could the authors show the vertical aerosol dry mass distribution with P-3B measurement? This information helps to untangle the compounding impacts of aerosol hygroscopicity and aerosol dry mass on aerosol extinction (and backscattering).

**Vertical profiles of aerosol dry mass distribution with P-3B measurements show basically constant values with altitude. But the main limitation for aerosol mass measurements is that they require large integration time and typically data are provided every 15 min. That is why we cannot make more conclusions. We will add clarifications to this point in the conclusion section and in the description of Table 4.**

Technique corrections:

1. Page 8 line 239: Section number of "computations of aerosol radiative forcing and heating rates" should be 2.4 instead of 2.3.

2. Page 9 line 264: change "hygrosocpicity" to "hygroscopicity".

3. Page 12 line 1: Replace "the use of" with "the".

4. Page 20 line 599: Please change "imaginary refractive index" to "real refractive index".

5. Page 31 line 884: change first "July 29" to be "July 22".

6. Page 31 line 885: delete the second "July" in "July 29 July".

7. Page 32 line 902: change "July 29" to be "July 22".

8. Page 32 line 919: missing "." after the sentence.

**We thank the referee for his/her time to detect all these typos. We will accordingly correct in the revised manuscript.**